# Systematic review of indoor residual spray efficacy and effectiveness against *Plasmodium falciparum* in Africa

Ellie Sherrard-Smith[1], Jamie T. Griffin[2], Peter Winskill[1], Vincent Corbel[3], Cédric Pennetier[3,4], Armel Djénontin[5], Sarah Moore[6,7,8], Jason H. Richardson[9], Pie Müller [6,7], Constant Edi[10], Natacha Protopopoff[11], Richard Oxborough[12], Fiacre Agossa[13], Raphael N'Guessan[11], Mark Rowland[11] & Thomas S. Churcher [1]

Indoor residual spraying (IRS) is an important part of malaria control. There is a growing list of insecticide classes; pyrethroids remain the principal insecticide used in bednets but recently, novel non-pyrethroid IRS products, with contrasting impacts, have been introduced. There is an urgent need to better assess product efficacy to help decision makers choose effective and relevant tools for mosquito control. Here we use experimental hut trial data to characterise the entomological efficacy of widely-used, novel IRS insecticides. We quantify their impact against pyrethroid-resistant mosquitoes and use a *Plasmodium falciparum* transmission model to predict the public health impact of different IRS insecticides. We report that long-lasting IRS formulations substantially reduce malaria, though their benefit over cheaper, shorter-lived formulations depends on local factors including bednet use, seasonality, endemicity and pyrethroid resistance status of local mosquito populations. We provide a framework to help decision makers evaluate IRS product effectiveness.

[1] MRC Centre for Global Infectious Disease Analysis, Department of Infectious Disease Epidemiology, Imperial College London, Norfolk Place, London W2 1PG, UK. [2] School of Mathematical Sciences, Queen Mary University of London, Mile End Road, London E1 4NS, UK. [3] Institut de Recherche pour le Développement (IRD), Maladies Infectieuses et Vecteurs, Ecologie, Génétique, Evolution et Contrôle (MIVEGEC), University of Montpellier, 34393 Montpellier Cedex 5, France. [4] Institut Pierre Richet, BP1500 Bouaké, Côte d'Ivoire. [5] Faculté des Sciences et Techniques, Université d'Abomey-Calavi, Cotonou, Benin. [6] Epidemiology and Public Health Department, Swiss Tropical and Public Health Institute, Socinstrasse 57, PO Box, 4002 Basel, Switzerland. [7] University of Basel, Petersplatz 1, 4001 Basel, Switzerland. [8] Ifakara Health Institute, Bagamoyo Research and Training Centre, Bagamoyo, Pwani, Tanzania. [9] Innovative Vector Control Consortium, Pembroke Place, Liverpool L3 5QA, UK. [10] Centre Suisse de Recherches Scientifiques en Cote d'Ivoire, Abidjan 01, BP 1303 Abidjan, Côte d'Ivoire. [11] Department of Disease Control, Faculty of Infectious and Tropical Diseases, London School of Hygiene and Tropical Medicine, Keppel Street, London WC1E 7HT, UK. [12] PMI VectorLink Project, Abt Associates, 6130 Executive Boulevard, Rockville, MD 20852, USA. [13] Centre de Recherche Entomologique de Cotonou (CREC), Cotonou, Benin. Correspondence and requests for materials should be addressed to E.S.-S. (email: e.sherrard-smith@imperial.ac.uk)

The mass distribution of long-lasting insecticide treated nets (LLINs) and the spraying of residual insecticides on indoor surfaces (indoor residual spraying, IRS) are together estimated to have averted 517 million cases from 2000 to 2015[1]. LLINs are attributed with the majority of this success, although bednets alone will be insufficient to push the parasite to elimination[2,3]. LLINs, IRS and the use of prophylactic or curative drugs in areas of high endemicity are the only widely used tools for preventing malaria[4]. Whilst these are proven technologies when used individually, their combined benefit is generally poorly understood[5]. The few studies that have considered these tools in combination give seemingly contradictory results[4,6] (likely due to location-specific factors that impact IRS and LLIN effectiveness) although the impact can be clear[7,8].

The substantial decade-long reduction in global malaria burden stalled in 2016 with an estimated increase of 5 million cases[9]. Given the reliance of global malaria control on LLINs there are fears that recent advances in control are threatened by the emergence of mosquitoes that are resistant to pyrethroids[10–12]. Yet measuring the public health impact of pyrethroid resistance is challenging[13,14]. Pyrethroids are currently the only widely-distributed class of insecticide on LLINs but wild mosquitoes are increasingly able to survive pyrethroid exposure[15]. As a potential solution, next-generation bednets and IRS products are being developed. Piperonyl butoxide (PBO) is a synergist that inhibits specific metabolic enzymes within mosquitoes that can detoxify pyrethroids, and has been incorporated into pyrethroid-treated LLINs to increase the insecticidal effect[16]. A recent randomised control trial (RCT) demonstrated PBO LLINs had a substantially bigger public health impact than pyrethroid-only LLINs in the context of pyrethroid resistance[8]. New, dual active ingredient (AI) nets (for example refs. [17,]18) combine a pyrethroid with a secondary chemistry that has an alternative mechanism of action. Nets remain essential for personal protection[19], however, until new non-pyrethroid LLINs become available, any loss in their ability to kill or deter mosquitoes is predicted to reduce the public health benefit for the community in pyrethroid resistance settings[20].

Pyrethroids have also been used ubiquitously for IRS[21] although most countries have transitioned to insecticides with alternative modes of action following the emergence of pyrethroid resistance. IRS can have significant public health value[22] however barriers to its use, including cost, logistics, acceptance and the lack of clear evidence of impact[23], limit its wider adoption. Combining pyrethroid-LLINs with pyrethroid-IRS risks exacerbating the spread of pyrethroid resistance. Therefore, the World Health Organization (WHO) recommends that, additional to LLINs, National Malaria Control Programmes (NMCPs) should introduce focal IRS with non-pyrethroids, and pre-emptively rotate active ingredients to slow the emergence of resistance[24]. Achieving this requires the development of multiple IRS products with different insecticidal modes of action[25,26].

Several IRS products have recently become available and others, currently under review by WHO, are likely to launch soon. Available products include: (i) Actellic®300CS (Syngenta), a micro-capsule suspension formulation of pirimiphos-methyl (an organophosphate); (ii) SumiShield®50WG (Sumitomo Chemical), a new IRS product containing clothianidin (a neonicotinoid)[27] (http://www.who.int/pq-vector-control/prequalified-lists/who_dec_doc_sumishield50wg.pdf?ua=1). There are currently five other non-pyrethroid WHO-recommended insecticides for IRS against malaria vectors: DDT, malathion, fenitrothion, bendiocarb (sometimes sprayed biannually) and propoxur (http://www.who.int/whopes/insecticides_irs_2_march_2015.pdf?ua=1).

IRS product effectiveness varies depending on factors including: (i) impact on mosquito populations (for example, an ability to kill or deter mosquitoes from entering a sprayed structure); (ii) impact duration (the residual half-life); (iii) where and when sprays are deployed (local malaria endemicity, seasonality of transmission and timing of IRS, mosquito species, human behaviour and net-use), and; (iv) spray quality and coverage. Which compounds to choose in a location and how best to rotate products, is further complicated by product price which can range from USD 2–3 to roughly USD 20 per unit[28] (a unit is standardised across products to cover approximately 250 m$^2$ of wall surface). Whilst long-lasting insecticides may be more effective, relatively shorter-acting but cheaper compounds might be appropriate in areas with short transmission seasons. To determine the additional benefit of adding IRS in a location with a specified level of pyrethroid resistance requires the rigorous characterisation of the efficacy of different IRS compounds on pyrethroid resistant vector populations and an understanding of the loss of efficacy of pyrethroids in areas where resistance has developed.

The WHO requires novel vector control products falling outside an established intervention class to provide epidemiological evidence of public health impact, typically through community RCTs[29] (http://www.who.int/neglected_diseases/vector_ecology/VCAG/en/). Novel products falling within an existing intervention class need not show epidemiological impact but instead can be recommended based on safety, quality and entomological efficacy data alone[30]. Experimental hut trials are the standard method for assessing the entomological impact of new IRS compounds[31] although they provide no direct information on the products' public health impact. Dynamic transmission models can be used in compliment to make public health predictions from experimental hut trial data[20,32–36]. The evidence-base for the use of transmission dynamics models in predicting vector control intervention effectiveness is currently lacking. The marked difference in IRS product efficacy in settings with different mosquito populations, and the recent proliferation of new IRS products, warrants its further investigation as conducting multiple RCTs for each product is financially challenging.

Here, experimental hut data are systematically assessed to characterise different IRS product efficacies against anopheline mosquitoes. We statistically assess IRS impact on mosquito mortality, blood-feeding and deterrence (whether mosquitoes preferentially enter unsprayed over sprayed structures) and how these impacts vary temporally. The impact of pyrethroid resistance on pyrethroid-IRS (approximated using the percentage of mosquitoes that survive during exposure to a diagnostic dose of a pyrethroid chemistry in 24-h WHO bioassay susceptibility tests) is also quantified statistically. A widely-used transmission dynamics mathematical model[2] is then employed to predict the public health impact of IRS with varying insecticides in areas with different levels of LLIN coverage and pyrethroid resistance. The application of these models is demonstrated by comparing model predictions to a measured change in prevalence for a defined age-group, assessed by cross-sectional surveys in RCTs. A theoretical framework is provided to help decision makers evaluate IRS cost-effectiveness in specific settings.

## Results

**Initial efficacy.** The meta-analyses (Supplementary Fig. 1) identified 98 individual experimental hut trials reporting an initial efficacy (against 24-h mosquito mortality, blood-feeding inhibition, exiting or deterrence) from 25 published studies and a further 3 unpublished datasets (Supplementary Table 1, Supplementary Data 1, analysis 1). The WHO recommends that Phase II studies reporting experimental hut trial data include 24-h

product-induced: mortality, blood-feeding inhibition, exophily and deterrence[31] although more recently, some active ingredients may take longer to kill e.g. clothianidin. These key indicators are summarised at the earliest time point noted for each study (no more than 2 months since spraying) to minimise under-estimating the effects of shorter-duration products (Table 1). Absolute values of mortality, blood-feeding and exophily are presented (instead of insecticide-induced estimates, corrected by untreated control huts) to allow different studies to be rigorously combined accounting for different covariates and weighted studies according to the number of mosquitoes caught.

There is substantial variation in the initial efficacy estimates for all IRS active ingredients. The binomial logistic regression model indicated that on average organophosphates killed a greater proportion of mosquitoes relative to all other active ingredients in the first 2 months after spraying (Fig. 1a). More mosquitoes were predicted to die in West African huts relative to East African huts and fewer mosquitoes were killed on mud substrate compared to cement (Fig. 1a, Supplementary Table 2). A greater proportion of mosquitoes exited from huts sprayed with carbamate or pyrethroid compared to organophosphates or neonicotinoids. Exit traps in East African huts had proportionally more

**Table 1 Summary data on IRS product entomological impact**

| Insecticide chemical class | Indoor residual spraying IRS insecticide induced: | | | | |
|---|---|---|---|---|---|
| | N | Mean 24-h mortality %, (range) | Mean 24-h exiting %, (range) | Mean 24-h blood-feeding inhibition %, (range) | Mean 24-h deterrence %, (range) |
| Organophosphates | 41 | 87.6 (39.8–100) | 15.9 (−175 to 96.3) | 7.1 (−18.0 to 49.4) | 42.3 (−106.9 to 94.9) |
| Neonicotinoids[a] | 8 | 68.0 (44.5–100) | 9.5 (−37.6 to 49.6) | −4.2 (−22.9 to 20.4) | 18.0 (−59.1 to 70.0) |
| Carbamates | 7 | 77.8 (27.3–100) | 34.5 (0.3 to 100) | 11.9 (−29.4 to 84.7) | 30.9 (−12.6 to 90.9) |
| Pyrethroids | 30 | 44.3 (5.0–92.8) | 43.7 (−5.1 to 96.4) | 18.2 (−29.7 to 82.6) | 5.6 (−151 to 77.4) |
| Organochlorines | 5 | 53.8 (18.1–79.6) | 13.9 (−8.1 to 27.9) | 44.0 (6.0 to 81.3) | −18.3 (−77.0 to 39.6) |
| Pyrroles | 5 | 57.8 (49.4–71.0) | −12.3 (−100 to 59.2) | −11.0 (−58.4 to 39.9) | −46.2 (−237 to 51.9) |

Summary data for the initial efficacy (within 2 months since spraying) for 24-h insecticide-induced mortality, exiting, blood-feeding inhibition and deterrence for different indoor residual spray (IRS) chemical classes (Organophosphates: pirimiphos-methyl (Actellic® 300CS), fenitrothion; Neonicotinoids: clothianidin (SumiShield® 50WG); Carbamates: bendiocarb; Pyrethroids: alphacypermethrin, lambda-cyhalothrin, permethrin, deltamethrin; Organochlorines: DDT (dichloro-diphenyl-trichloroethane) and; Pyrroles: chlorfenapyr) as evaluated in experimental hut trials. The full dataset is provided in Supplementary Data 1
[a]Adjusted 72-h mortality presented, see Methods

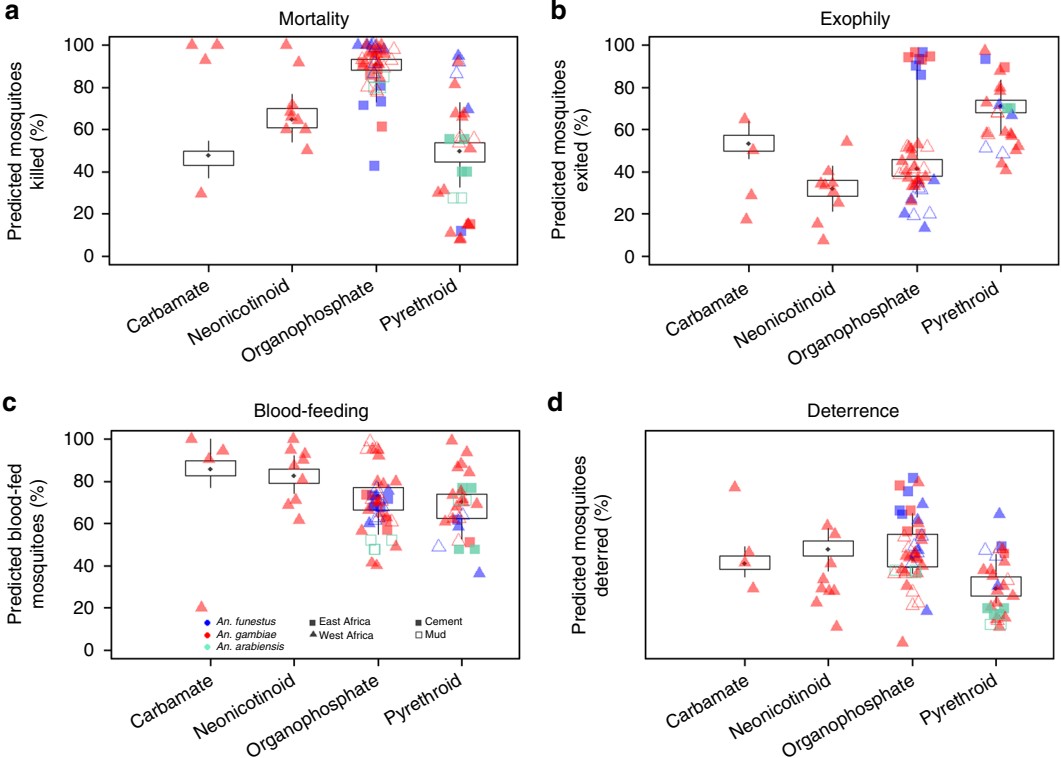

**Fig. 1** Summary estimates for the level of mosquito mortality (**a**), exophily (**b**), blood-feeding (**c**) and insecticide-induced deterrence (**d**) as assessed in experimental hut trials with different indoor residual spray chemical classes measured within 2 months of spraying. Black box-plot show the binomial logistic model predictions (median (dark point), 25th and 75th uncertainty intervals indicated by box, 5th and 95th uncertainty intervals indicated by whiskers) which are weighted for the number of mosquitoes caught in experimental huts. The symbols show the raw data (also provided in Supplementary Data 1, analysis 1) and are classified according to the type of experimental hut (shape of symbol) and the hut substrate (symbol fill) as noted in the key in **c**. Point colour indicates the mosquito species complex, be it blue (*A. funestus s.l.*) or red (*A. gambiae s.l.*). Mosquitoes in the *A. gambiae s.l.* complex which were identified as *A. arabiensis* are shown in green. Supplementary Fig. 2–5 show the Bayesian posterior predictive fits against disaggregated data

mosquitoes than West African huts, and both *Anopheles funestus* and *Anopheles gambiae s.l.* were predicted to be in exit traps more than *Anopheles arabiensis* (though there were only 5 data sets with *A. arabiensis* mosquitoes, Fig. 1b). Blood-feeding was greater in huts sprayed with carbamate relative to huts sprayed with other chemistries, *A. gambiae s.l.* was predicted to have blood-fed more than *A. arabiensis* and blood-feeding was greater in West African huts relative to East African huts (Fig. 1c). There were minimal differences in the degree to which products deterred mosquitoes (Fig. 1d). Further summary statistics of these initial impacts of IRS chemistries are provided in Supplementary Methods (Supplementary Table 2).

**Temporal characterisation.** The public health impact of different insecticides will depend on how long they last relative to the length of the transmission season. Studies identified in the meta-analysis reporting the number of mosquitoes killed, blood-fed or exited at 3 different time-points or more were collated (Table 2, Supplementary Data 1, analysis 2). There was no significant

difference in the initial efficacy estimated for analysis 1 and those which provide sufficient temporal information to be used in analysis 2, for neither 24-h induced mortality, blood-feeding inhibition, exiting nor deterrence (generalised linear models $p >$ 0.1 in all instances). This smaller dataset (number of studies ($N$) = 12 listed in Table 2, providing 28 sets of time series data) is used to characterise how mosquito mortality and blood-feeding changes over time for each IRS active ingredient assessed (Fig. 2). The change in the deterrent action of IRS over time cannot be rigorously evaluated in standard experimental hut trials (see Methods section) so here only the initial impact is assessed with the rate of decay following that observed with mosquito mortality (Fig. 2—row 3). The mean proportion of mosquitoes killed decreases with time since spraying, whilst the proportion of mosquitoes successfully blood-feeding increases. Efficacy varies substantially both within and between the different IRS active ingredients tested (Fig. 2, Supplementary Fig. 6). Generally, experimental hut data for both Actellic® 300CS and SumiShield® 50WG indicated these products induce high mortality over a prolonged period relative to the pyrethroids (given the average

**Table 2 Data available to assess temporal changes in IRS efficacy**

| Study | Location | Type of hut | Wall surface | Where was the IRS applied? | Sleepers and nets | Mosquito species present | Insecticides tested |
|---|---|---|---|---|---|---|---|
| | | **Data from published studies** | | | | | |
| 1[63] | Benin | West African | Cement | Walls (not ceiling) | No nets | *A. gambiae s.l.* | Pyrethroids (lambda-cyhalothrin), Actellic® 300CS, Bendiocarb |
| 2[64b] | Benin | West African | Cement | Walls (not ceiling) | No nets | *A. gambiae s.l.* | Pyrethroids (alphacypermethrin [2[a]], deltamethrin [14[a]], lambda-cyhalothrin[15[a]]), |
| 3[17] | Benin | West African | Cement | Walls and ceiling | No nets | *A. gambiae s.l.* (*A. coluzzi* and *gambiae ss*) | Pyrethroids (alphacypermethrin), |
| 4[25b] | Benin | West African | Cement [4[a]] and mud [13[a]] | Walls and ceiling | No nets | *A. gambiae s.l.* | Pyrethroids (lambda-cyhalothrin), Actellic® 300CS, |
| 5[65b] | Cote D'Ivoire | West African | Cement | Walls and ceiling | No nets | *A. gambiae s.l.*, *A. funestus* | Pyrethroids (lambda-cyhalothrin), Actellic® 300CS |
| 6[66b] | Benin | West African | Cement | Walls and ceiling | Untreated nets | *A. gambiae s.l.* | Pyrethroids (deltamethrin [6[a]], alphacypermethrin [16[a]]), Bendiocarb [6[a]] |
| 7[67b] | Benin | West African | Cement [7], mud [17], plywood [18] | Walls and ceiling | No nets | *A. gambiae s.l.* | Pyrethroids (3 x deltamethrin [7[a], 17[a], 18[a]]), Clothianidin (200 mg m$^{-2}$) [7[a]] |
| 8[39] | Burkina Faso | West African | Unknown | Walls (not ceiling) | Holed nets | *A. gambiae s.l.* | Bendiocarb |
| 9[26] | Tanzania | East African | Mud | Walls and ceiling | No nets | *A. arabiensis* | Actellic® 300CS |
| 10[68] | Benin | West African | Cement | | No nets | *A. gambiae s.l.* | Pyrethroids (deltamethrin), Sumishield® 50WG |
| | | **Additional data from unpublished studies** | | | | | |
| 11 unpublished data 1 Vincent Corbel | Benin | West African | Concrete | Walls and ceiling | No nets | *A. gambiae s.l.* (*A. gambiae s.s.* 95%; *A. arabiensis* 5%) | Pyrethroids (deltamethrin), Bendiocarb, SumiShield® 50WG |
| 12 unpublished data 2 Pie Müller | Cote D'Ivoire | West African | Cement (plywood ceilings) | Walls and ceiling | Untreated, holed nets | *A. gambiae s.l.* | Actellic® 300CS, SumiShield® 50WG |

Data available to assess temporal changes in IRS efficacy (against mortality, successful blood-feeding and deterrence) of mosquitoes in free-flying experimental hut trials of different indoor residual spray compounds
[a]Number refers to the coded symbols in Figs. 2 and 3
[b]Multiple time series for an IRS product, see Supplementary Data 1, analysis 3b

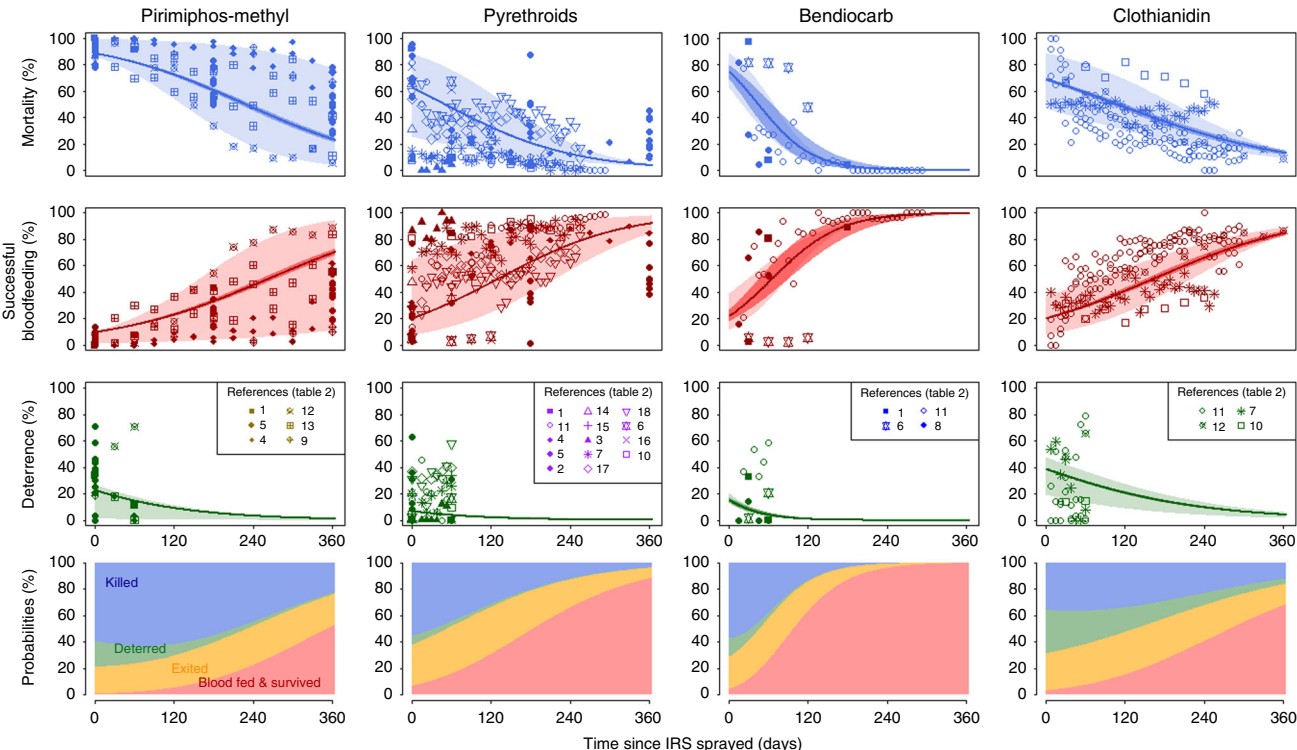

**Fig. 2** Summary of the temporal entomological impact of different IRS compounds. Probability of mosquitoes dying (top row), successfully blood-feeding (surviving and feeding) (row 2) or being deterred (row 3) in experimental hut trials over time. Row 4 summarises the best fit probability outcomes per feeding attempt for a mosquito to successfully blood-feed (red), exit without feeding (orange), be deterred before entering (green) or be killed (blue) for the different IRS products; pirimiphos-methyl: Actellic®300CS (column 1), pyrethroids: lambda-cyhalothrin, deltamethrin and alpha-cyhalothrin (column 2), bendiocarb: bendiocarb (1 spray round per year) (column 3) and neonicotinoids: clothianidin, SumiShield®50WG (column 4). Symbol shapes indicate the different studies (legend key references study numbers in Table 2 corresponding to 1[63], 2, 14 and 15[64], 3[17], 4 and 13[25], 5[65], 6 and 16[66], 7, 17 and 18[67], 8[39], 9[26], 10[68], 11 and 12 are previously unpublished data). Solid lines indicate the best fit statistical model to the mean data, weighted by sample size in different studies, and the dark-shaded area shows the 90% credible intervals around these best fit lines. The maximum and minimum data for each unique time point, for each IRS product, are fitted to capture the uncertainty in predicted performance of IRS products over time, these ranges are shown as pale polygons in rows 1–3 for each product. There is much uncertainty in the measurement for deterrence (row 3) because huts testing products that are sprayed onto walls cannot be easily rotated, we therefore simply fit to the initial deterrence measured and consider the depreciation of the deterrence effect to match that of mortality (further detailed in Supplementary Methods). Supplementary Fig. 6 shows individual study fits for these data

level of pyrethroid impact in experimental hut trials is estimated from studies in locations that may have pyrethroid resistance already) and bendiocarb. Bendiocarb initially induced high mortality (above 60%) but this declines rapidly. There is considerable variability in the temporal trends of the same product between studies, though it is unclear whether this specifically reflects differences between the local mosquito population, procedural or other factors. Using these data, the probability outcomes of a mosquito feeding attempt can be determined across time (Fig. 2—row 4). Parameter sets to describe these fits are provided in Supplementary Table 3.

**Pyrethroid resistance**. The impact of reduced susceptibility of local mosquitoes to pyrethroid insecticides on the efficacy of pyrethroid-IRS is summarised in Fig. 3 (Supplementary Data 1, analysis 3). Essentially, the initial efficacy of the pyrethroid-IRS is reduced and the active life-length of the insecticide is shorter as fewer mosquitoes are susceptible to a pyrethroid-IRS. Pyrethroid resistance is measured using a discriminatory dose bioassay test[37]. There is a strong association between the level of resistance and 24-h mosquito mortality observed in the experimental hut trial (Fig. 3a), matching a similar trend seen with LLINs[20]. For mosquitoes that enter the hut there is a clear increase in successful blood-feeding with increasing mosquito survivorship, as those that previously fed and then died, later survive (Fig. 3b). The level

of deterrence initially observed in an experimental hut trial also decreases with increasing survivorship, though there is more uncertainty about this rate of decline (Fig. 3c). Insecticide resistance also diminishes the duration of the killing ability of the active ingredient for the pyrethroid-IRS. We represent this showing that the time taken to kill 50% of the mosquitoes (lethal time $LT_{50}$) is reduced in experimental hut trials given increased mosquito survival during bioassay testing (Fig. 3d). The same is true for $LT_{20}$, and $LT_{80}$. A summary of the combined impact of resistance on the probability outcome of each mosquito feeding attempt is shown in Fig. 3e. Fewer mosquitoes are killed, more mosquitoes blood-feed and fewer are deterred from sprayed huts as pyrethroid resistance increases immediately after spraying. However, some mosquito mortality due to IRS is predicted even when all mosquitoes are surviving the discriminatory dose bioassay because there is measurable mortality in hut trials at $t = 1$ day at this level of resistance (Fig. 3a). Similar trends are also seen in resistant populations over time since spraying (Fig. 3f).

**Comparing model predictions and randomised control trials**. Transmission dynamics models provide a means of converting entomological measures of IRS efficacy into a prediction of their impacts on public health. To illustrate the utility of the IRS characterisation, a transmission dynamics model is used to predict the outcome of two recent IRS RCTs. Standard pyrethroid

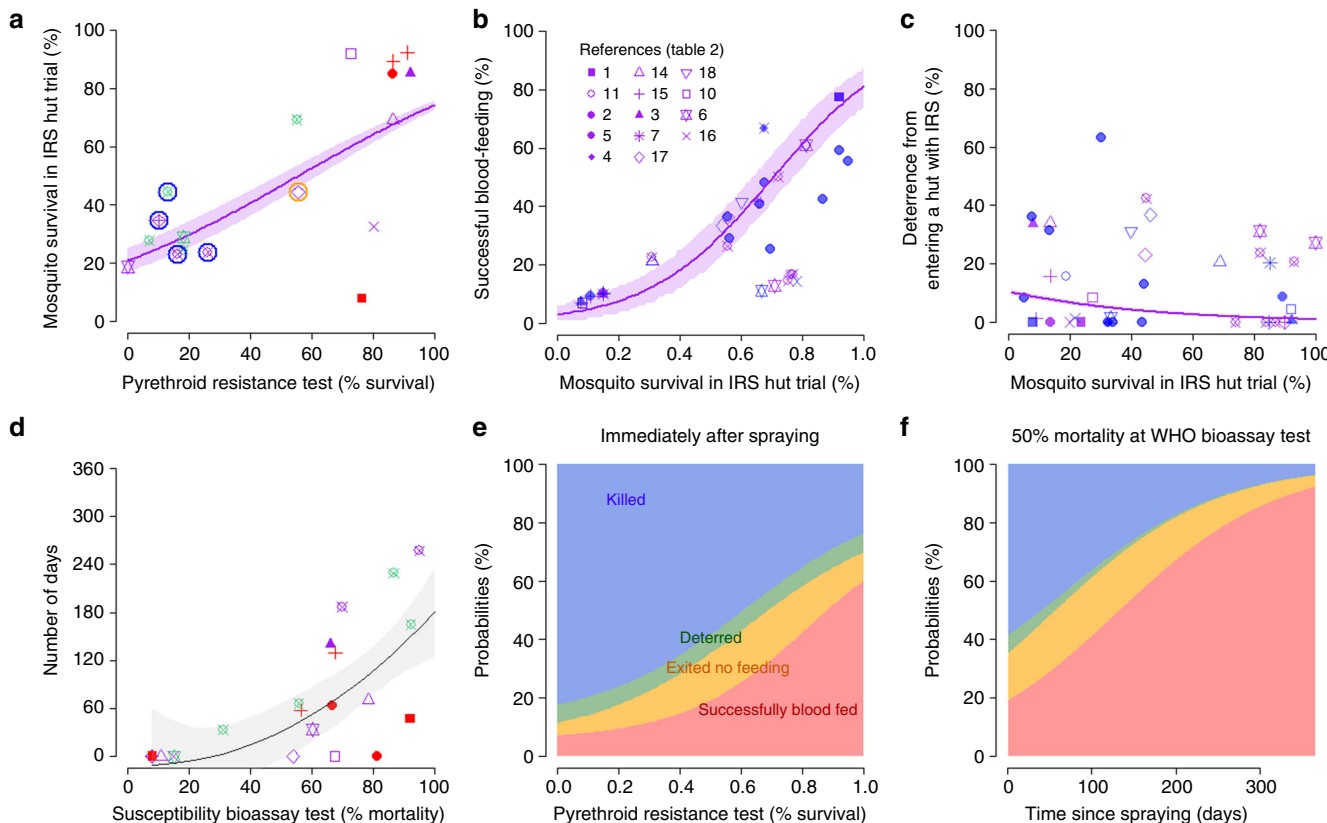

**Fig. 3** The impact of pyrethroid resistant mosquitoes on the efficacy of pyrethroid-IRS. **a** The association between pyrethroid resistance in a mosquito population (measured as percentage survival over 24-h after a 60-min-exposure to a standard dose pyrethroid in the bioassay test) and 24-h mosquito survival immediately (time $t = 1$ day) after IRS spraying in an experimental hut trial (Supplementary Data 1, analysis 3a). Data represent *A. gambiae s.l.* complex (no circle), *A. funestus s.l.* data (symbol circled in blue) or *A. arabiensis* (circled in orange). The pyrethroid active ingredient tested in the bioassay either matched (purple) or mis-matched (red) the pyrethroid active ingredient used in the hut trial, or bioassay data was taken from a second study reported at the time of the IRS hut trial (green) (Supplementary Data 1, analysis 3a). The relationship between 24-h mosquito survival in a standard pyrethroid hut trial and the probability that a mosquito, on entering a hut, will successfully blood-feed (**b**) or preferentially enter a hut (**c**) without IRS (deterrence). In **b** and **c**, the 18 data from 13 studies with standard pyrethroid discriminating dose bioassay data (purple) are shown together with the 21 data from 11 studies for pyrethroid-IRS with time series but no bioassay measurement (blue) (Supplementary Data 1, analysis 3). Any symbol not noted in the key is included in addition to studies listed in Table 2 within Supplementary Data 1, analysis 3b. **d** The relationship between mosquito survival and longevity of the IRS. The time taken in days until less than 50% of mosquitoes die within 24-h (*y*-axis); the longer this duration, the longer the activity of the IRS (see Methods). **e** Summary of how pyrethroid resistance is predicted to influence the probability that a host-seeking mosquito will be killed (blue), deterred from entering (green), exit without feeding (orange) or successfully feed and survive (red) during a single feeding attempt in a hut freshly sprayed with pyrethroid-IRS ($t = 1$ day). Panel **f** shows the 3D relationship for pyrethroid resistance (50% survival at bioassay), mosquito outcome (colours as per panel **e**) and time since spraying

LLINs were distributed to participants of all arms of the trial analysed here so their efficacy was adjusted for the impact of pyrethroid resistance[20]. Overall best fit model predictions broadly match observed data for a single round of Actellic® 300CS[8] (Fig. 4a) or bendiocarb[38] (Fig. 4b). The uncertainty in Actellic® 300CS efficacy has a relatively minor impact on the uncertainty of public health predictions for the first 10 months after application but substantial variation after this point. A similar pattern is seen in the bendiocarb data though the uncertainty manifests itself earlier due to its shorter residual activity. Predictions of the impact of IRS from each of the individual studies (Supplementary Fig. 6) are shown as thin lines demonstrating the contrasting predictions that are determined from these independent parameter sets (individual study parameters are provided in Supplementary Table 4). Interestingly, the Actellic® 300CS data on *A. gambiae s.l.*, the most prolific mosquito present in Muleba district[8] more closely predict the observed change in prevalence during the trial. Both mud and cement are used as wall substrate for houses in Muleba and the experimental hut data for

these surfaces[25] most closely reflects the measured impact. There is less uncertainty in the predicted impact of bendiocarb (Fig. 4b) though the efficacy model was parameterised with fewer experimental hut studies which had less variability (i.e. they were all from West African design, *A. gambiae s.l.* mosquitoes and probably cement substrate, though this was unknown for one study[39]).

**Predicting the public health impact of IRS**. The public health benefit of IRS with different compounds is then predicted using the model for a wider range of settings and LLIN use. Long-lasting products (Actellic® 300CS and SumiShield® 50WG grouped together) used at 80% coverage, are estimated to avert up to 500 clinical cases per 1000 people per year in perennial settings with moderate endemicity, when the level of pyrethroid resistance is very high and bednet use is low (Fig. 5a–c). The uncertainty in these estimates is illustrated in Supplementary Fig. 7 which show predictions using parameters that define the best- and

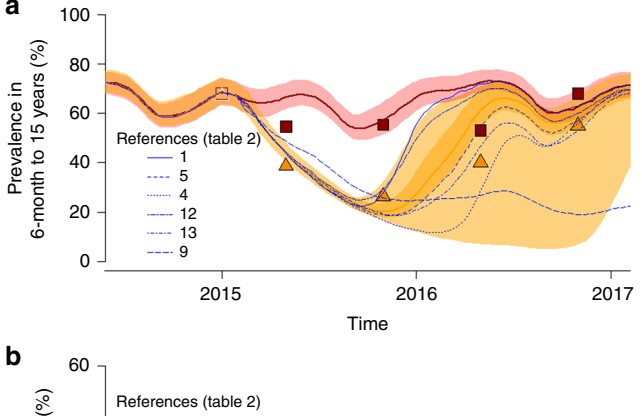

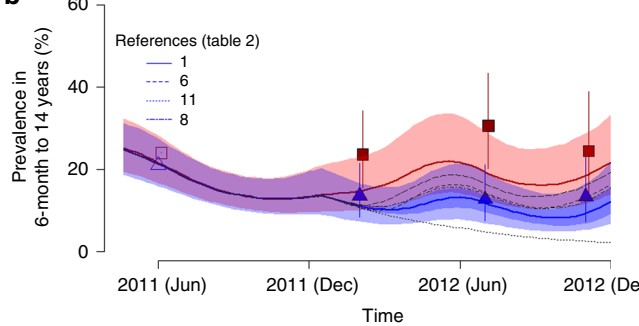

**Fig. 4** Comparison of the predicted impact of IRS on malaria prevalence compared to that measured in randomised control trials. **a** Comparison of best fit standard LLINs (solid red line) vs standard LLINs + a single round of Actellic® 300CS (solid orange line)[8]. Shaded area indicates uncertainty in model predictions. The paler shaded area around the IRS lines shows additional uncertainty driven by variability in IRS efficacy (as illustrated in Fig. 2, Supplementary Table 3). The thin grey lines, noted in the key, denote IRS predictions parameterised separately for the individual experimental hut studies (Supplementary Table 4). **b** Comparison of standard LLINs (solid red line) vs standard LLINs + two rounds of bendiocarb (solid blue line)[38]. All models were parameterised using the data listed in Table 3 and estimates of the seasonality of transmission within the Kagera district and fitted to baseline prevalence (open symbols). Predictions of any change in prevalence for the respective age cohort measured were then made. The observed estimates for prevalence obtained during cross-sectional surveys of each RCT are plotted as closed symbols (in **b** vertical lines indicating 95% confidence intervals reported in the RCT[38])

worst- performing experimental hut trials. In highly seasonal settings, short-lasting IRS products (such as bendiocarb sprayed once at an optimal time prior to the transmission season) can span the duration of relatively short transmission seasons and avert similar numbers of cases as long-lasting products over the course of a single short season (Fig. 5d–f). Bendiocarb is often sprayed biannually so we also predicted the cases averted for the biannual scenario for this insecticide which demonstrates bendiocarb can be as effective as long-lasting IRS. This would need to be considered for cost-effectiveness estimates, although the product cost per unit may be low, the logistical costs of implementing an additional spray campaign remain high. The cases averted from using IRS is greater at higher levels of resistance and lower net coverage levels because the IRS is then able to mitigate lost impact from the pyrethroid-LLINs (Fig. 5). Ultimately, the benefits of adding IRS will be location specific and dependent on multiple factors including the local level of endemicity, mosquito species, house wall substrate, length of the transmission season and existing LLIN coverage and use.

To provide a framework for decision makers, the relative efficacy (relative reduction in clinical cases due to using IRS at 80% coverage vs no change in historic IRS use in a scenario with a

pre-defined level of resistance ranging from 0 to 100%) is estimated (Supplementary Data 2). Each administration subunit 1 across sub-Saharan Africa is predicted given assumptions that are made about the proportion of local mosquitoes of different species, local bednet coverage and historic net use (as estimated up to 2015[40]). To give an idea of the uncertainty in these estimates, Supplementary Data 2 also show the estimated cases averted if we use parameter estimates describing the experimental studies with the least or most impact on mosquito behaviours for each IRS active ingredient. In places already using high net coverage the additional benefit of IRS is relatively low whereas where bednets are not implemented or used at lower coverage, IRS is predicted to have a big impact. Short-lasting bendiocarb IRS estimates are provided for either annual or biannual application of the active ingredient in Supplementary Data 2. This will enable local decision makers to take cost data and predict the most cost-effective option in their location depending on available funds and programme goals.

## Discussion

The use of IRS to supplement LLINs for malaria control and elimination is increasing in part due to concerns of pyrethroid resistant mosquitoes impeding bednet efficacy and the drive for malaria elimination. This modelling exercise highlights that the added public health benefit of the WHO policy to add IRS to LLINs can be substantial in areas where bednet usage is low and pyrethroid resistance is a concern. However, the scale of the impact varies according to the type of insecticide sprayed and where it is used. These results are in broad agreement with a recent epidemiological literature review on combining IRS and LLIN interventions in Zambesia, Mozambique and Bioko, Equatorial Guinea[41]. A further study in Burundi found no additional impact on prevalence when LLINs were combined with pyrethroid-IRS[42]. An RCT in Southern Benin also showed no additional epidemiological benefit of annual bendiocarb-IRS over LLINs alone[6] although our analyses show that the residual half-life of bendiocarb is relatively short (less than 2 months) compared to the long transmission season in Benin which may partially explain this lack of additional benefit. A key consideration for trial design is the timing of bednet re-distributions. Nets are generally very effective during the first year when LLIN usage is high, nets are untorn, and the active ingredient is most effective. Therefore, fewer mosquitoes blood feed and rest on the wall and have less contact with insecticide-sprayed surfaces. This can mask the impact of IRS (or other interventions) used on top of nets and may explain some of the discontinuities in IRS and LLIN RCT results.

The recent registration of Actellic® 300CS and SumiShield® 50WG means that for the first-time multiple long-lasting IRS products are available with different modes of action that achieve broadly equivalent reductions in malaria burden across Africa. It is therefore imperative that pre-emptive rotation of products or their use in mosaics is implemented to maintain the efficacy of both insecticides[24]. The decline in the efficacy and public health effectiveness of pyrethroid-IRS highlights the dangers of the use of interventions with single modes of action, especially given the ubiquity of pyrethroid-based LLINs across Africa. This work focuses on the impact of resistance as measured in experimental hut trials. Pyrethroid-IRS might still provide protection against mosquito populations no longer killed by the insecticide (and therefore not detected in a standard hut trial) as it may reduce their fecundity or elicit other sub-lethal effects[43–45] but such impacts may be minimal given the impacts seen when resistant areas switch to alternative IRS products[15,22]. Similarly the transmission model employed here assumes that mosquito biting times remain constant throughout the simulations and between

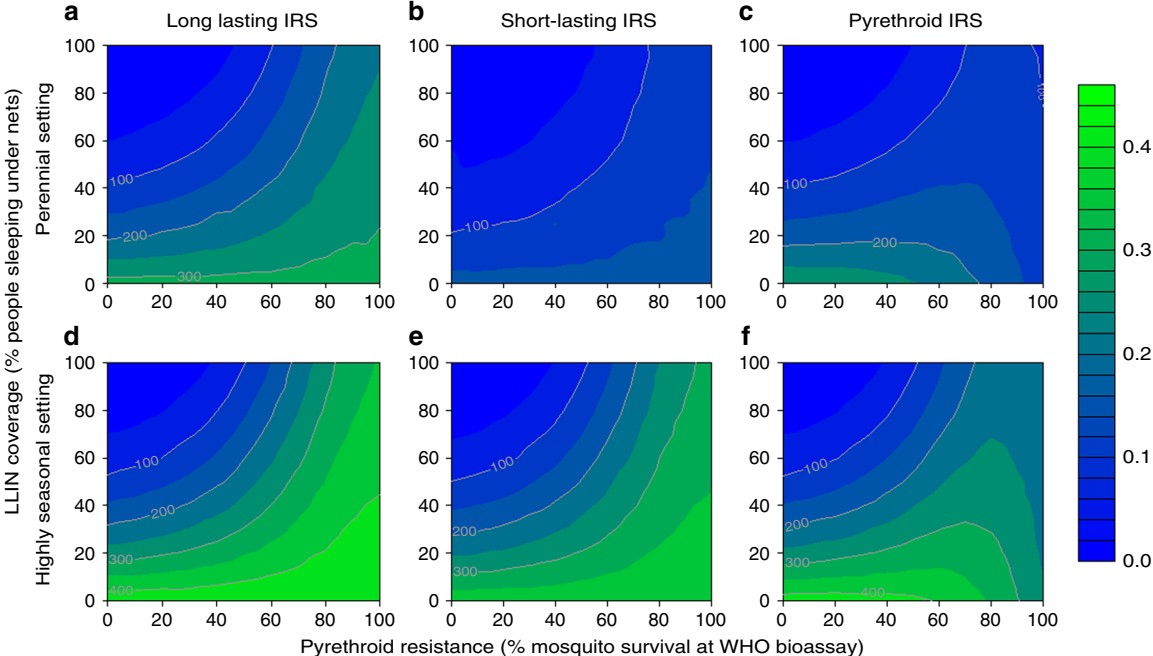

**Fig. 5** The additional impact of adding IRS to bed nets. The predicted number of malaria cases averted by annual application of IRS to a population with an existing level of bednet use (0–100% cover, y-axes) and a defined level of pyrethroid resistance (measured as percentage survival in a standard pyrethroid discriminating dose bioassay, x-axes). Clinical cases averted are measured per 1000 people per year, following standard LLIN distribution in a moderate endemicity area (30% prevalence in 2–10-year olds in the absence of interventions) with perennial transmission (**a–c**), highly seasonal transmission (**d-f**). In all panels IRS is applied, untargeted, to 80% of the population using either a long-lasting IRS product (for example Actellic®) (**a**, **d**), a short-acting IRS product (for example bendiocarb, applied annually) (**b**, **e**) or a pyrethroid-IRS product (for example deltamethrin (**c**, **f**)). Long-lasting products avert more cases though short-lasting products perform substantially better in highly seasonal settings

locations and does not reflect any additional impact of insecticides inducing behavioural changes to transmission[46,47]. There is some evidence that continued presence of insecticides may cause increased outdoor mosquito biting[48] or earlier evening and later morning biting[49,50] as well as changes in indoor resting behaviours[51]. The greater the overlap in activity time between mosquitoes biting and resting indoors and people being indoors, but not under bednets, the greater is the potential impact on IRS effectiveness[32].

There is substantial variation in the duration of action (induced mortality, inhibited blood-feeding or deterrence) between studies with the same product. Some of these differences are likely to be procedural differences (such as wall-type[52]) though others will reflect true differences in the behaviour and susceptibility of local mosquito populations. There were too few studies to investigate these covariates statistically using the temporal data although investigation of the initial data indicated differences in IRS efficacy for different mosquito species and wall substrates. Future work is needed to enable this greater realism to be included in predictions, particularly as pyrethroid resistance profiles might vary substantially between species[53]. One major difference was that IRS in West African design huts induced much higher mortalities than Ifakara huts which are a closer representation to the house type in that part of Tanzania. This has been documented previously[54,55], so Ifakara hut studies were excluded from the analysis. No hut design is likely to truly capture the interaction between mosquitoes and IRS sprayed houses. Nevertheless, results presented here do show that other hut types can broadly predict public health benefits. Given their increased importance more research is needed to understand how differences in design influence measurement of IRS and LLIN efficacy so results from distinct locations can be fully interpreted and extrapolated beyond the settings where trials were carried out.

Despite the substantial variation between studies the meta-analysis was able to capture average efficacy of Actellic® 300CS and bendiocarb which matched predictions from two RCTs. Generally, the predicted impact of each insecticide using individual studies (Fig. 4) was broadly able to recreate the RCTs with the exception of a study where the principal mosquito was *A. arabiensis*, rather than *A. funestus* or *A. gambiae s.s.* and experimental huts were East African design[26], rather than West African design (Table 2). Further work with a greater number of studies is needed to validate the models and IRS characterisation to ensure predictions are as accurate as possible. Nevertheless, the results support the WHO policy allowing new IRS products which demonstrate non-inferiority to existing products in that class to be recommended based on entomological data alone[56]. RCTs which assess public health outcomes of new IRS products will still be essential, especially for products which may induce delayed mortality or sub-lethal effects on mosquito populations to ensure the full efficacy is captured. The variability between studies highlights the value for multiple experimental hut trials in different settings and ecologies as predictions derived from single studies are highly variable.

There are some key limitations to the presented analyses. First, we have minimal data on how different malaria vectors will be affected by IRS and have consequently assumed the same probability outcomes for each mosquito species. The substrate of local housing also impacts IRS efficacy. There were too few experimental hut studies on each insecticide to reliably differentiate these effects for specific scenarios. Currently, to the best of our knowledge there are no published studies where experimental hut trials were conducted in the same location as an RCT. This will be important to ensure the efficacy of the interventions are being assessed against the same mosquito populations although we can broadly recreate RCT outcomes with the meta-analysis approach

outlined here (although there can be considerable uncertainty). Experimental hut data are often aggregated which means that assumptions on the proportion of mosquitoes that are feeding and surviving during the trials need to be made. Deterrence is notoriously challenging to measure and the assumption is made that the waning effect for deterrence mirrors that for mortality, although this needs to be verified. The discriminating dose bioassay test has inherent limitations for measuring the level of pyrethroid resistance in wild mosquito populations that are outlined above and previously[20]. Finally, we do not consider behavioural resistance in mosquito species that may render indoor vector control less effective. The proportion of mosquito bites received indoors is assumed to be consistent across different settings here.

Here we provide a comprehensive method to assess IRS products using experimental hut data and extrapolate their impact for public health outcomes. Model simulations indicate that the lost impact of pyrethroid-IRS and pyrethroid-LLINs in the presence of pyrethroid resistant mosquitoes can be mitigated using IRS products with different modes of action and that new long-lasting products such as Actellic® 300CS and SumiShield® 50WG can have substantial public health benefit especially in areas with perennial malaria transmission. A full cost-effectiveness analysis is beyond the scope of this study and is needed to help inform policy. The price of different IRS and LLIN products are continually changing making it hard for programme managers to justify procurement decisions. Here we provide estimates for the number of cases averted per 1000 people per year at increasing levels of pyrethroid resistance for every administrative 1 unit in Africa considering local seasonality and LLIN coverage (Supplementary Data 2). These estimates are determined using IRS at 80% cover which may not be financially achievable everywhere or with all products. Different insecticides and formulations have different effects on different wall surfaces[26] as well as contrasting smells or propensity to leave stains which affects acceptability[57]. The present analysis assessing the potential impact of IRS at different levels of pyrethroid resistance can contribute to decision making. From these predictions, the greatest added value to LLIN is in areas where LLIN usage is low and pyrethroid resistance is high. NMCPs can combine these data with local, up-to-date cost information to generate broad cost-effectiveness estimates for implementing different IRS campaigns on top of existing LLIN programmes given their unique entomological and epidemiological settings. As the number of novel malaria control interventions increases these locally tailored strategies can help to achieve local goals and push for malaria elimination.

## Methods

**Data collation.** A meta-analysis of IRS experimental hut trials is used to summarise measures of IRS efficacy. Whilst experimental hut trials cannot account for all of the effects of IRS alone[58] they provide a relatively standardised method to assess IRS efficacy and are considered the entomological equivalent of a Phase II trial[31]. They are also a pivotal part of the testing of new products and are required by WHO Prequalification[29] which enables products to be bought by international procurers for low-income countries.

**Data extrapolation and exclusion criteria.** The meta-analysis was conducted based on the PRISMA guidelines which highlight how best to perform systematic reviews for clinical trial data. Here, we are interested in count data for mosquitoes in Phase II studies over a time series of multiple months. Four search engines were used (Web of Knowledge, PubMed, JSTOR and Google Scholar) to identify relevant data resources. Policy teams and author's regularly conducting these studies were also contacted to access unpublished resources. A schematic of the process (Supplementary Fig. 1) and table noting the reasons for excluding studies are included in Supplementary Table 1. To the author's knowledge, there has been no previous published systematic meta-analysis on IRS compounds tested in experimental hut trials. Studies are limited to trials conducted in Africa (where the biggest burden of falciparum malaria is found) and to mosquito species belonging to the Anopheles family (vectors of the disease).

**Summary statistics.** Experimental hut studies typically report 24-h product-induced: mortality, blood-feeding inhibition, exophily and deterrence[31]. Here we present absolute values of mortality, blood-feeding and exophily as measured in the treated huts. This is to allow the results of different studies to be appropriately statistically combined, though each are presented individually, and insecticide-induced estimates can be calculated from Supplementary Data 1, analysis 1. There is relatively little variation in the level of mortality, blood-feeding and exophily observed in the control (unsprayed) huts in the studies examined here and this method is consistent with previous modelling efforts[20]. Summaries of each are described below.

(i) Mortality: The number of female mosquitoes found in the hut which are dead on collection or die within the next 24-h is denoted $D$. In the following equations, the subscript denotes whether the number dead (or other characteristic) was measured in the control (unsprayed hut = $C$) or the sprayed hut ($T$). If $N$ is the total number of female mosquitoes that were found in the hut or exit traps then,

$$\text{Mortality}(\%) = 100 \times \frac{D_T}{N_T} \qquad (1)$$

(ii) Exophily: Exophily is the propensity for mosquitoes to rest outdoors after feeding which can diminish the impact of IRS. It is calculated as the number of female mosquitoes in exit traps ($E$) compared to the sum of the number collected in the hut and exit traps ($N$) as,

$$\text{Exophily}(\%) = 100 \times \frac{E_T}{N_T}. \qquad (2)$$

(iii) Blood feeding: The number of mosquitoes that are blood fed which were collected in the hut and exit traps is denoted $B$ so the percentage blood fed in a sprayed hut is given by,

$$\text{Blood fed}(\%) = 100 \times \frac{B_T}{N_T}. \qquad (3)$$

(iv) Deterrence: Deterrence induced by IRS is defined as the reduction in the entry rate of mosquitoes into experimental huts with or without IRS,

$$\text{Deterrency}(\%) = 100 \times \frac{(N_C - N_T)}{N_C}. \qquad (4)$$

**Comparison of the initial impact of IRS.** The first analysis summarises and compares the initial impact of different IRS products. Data were restricted to initial timepoints collected within 2 months of IRS application as the active ingredient decays with time, so that averaging across the whole dataset may mis-represent the initial potency of IRS as studies had different durations. Statistical models were fit to generate overall estimates of the efficacy of the chemical class. These explanatory factors included the mosquito vectors (classified at the species complex level and species level where possible, i.e. *A. arabiensis*, *A. funestus s.l.* and *A. gambiae s.l.*), experimental hut type (West or East African design) and hut wall substrate (cement or mud) alongside the chemical class used for the IRS (carbamate, clothianidin, organophosphate and pyrethroid). Preliminary data exploration revealed that there were too few data to perform an extensive statistical test on all covariates. To overcome this a subset of the full database was generated by removing Ifakara hut studies, wall substrates that were not mud or cement and chemistries other than pyrethroids, organophosphates, carbamates or neonicotinoids. Binomial logistic regression models were fitted to the remaining count data ($N = 78$) to estimate the number of mosquitoes that were dead in 24-h, had exited, blood-fed or been deterred by the IRS product. The predicted value for the proportion of mosquitoes being killed, exiting, blood-fed or deterred is calculated as:

$$\pi_i = logit^{-1} ln \pi_i / (1 - \pi_i) = \frac{\exp(\beta_0 + \sum_h \beta_h X_{hi})}{(1 - \exp(\beta_0 + \sum_h \beta_h X_{hi}))} \qquad (5)$$

where $\pi_i$ is the estimated proportion for the $i$th data (e.g. the proportion of mosquitoes killed), $\beta_0$ is the intercept, the subscript $h$ denotes the covariate of interest (taking number of 1 to $H$) and $X_h$ is a matrix of explanatory factors (mosquito species, hut type, substrate and chemistry sprayed) with coefficients $\beta_h$[59]. Bayesian models were fitted using Hamiltonian Monte Carlo sampling methods[60,61]. Four chains were initialised to assess the convergence of 2000 iterations, the first 1000 of each were discarded as burn in. The posterior distributions of parameters (4000 iterations) and 90% Bayesian credible intervals were estimated, posterior checks were performed using ShinyStan (version 1.0.0)[62] and visually confirmed to fit the data (Supplementary Fig. 2–5).

**Temporal characterisation of different active ingredients.** Four insecticide active ingredients, pyrethroids (including deltamethrin, lambda-cyhalothrin and alpha-cypermethrin), pirimiphos methyl, bendiocarb and clothianidin were further characterised from data identified in the meta-analysis. These four groups of active ingredients were chosen as they are likely to be the main insecticides used by NMCPs for IRS in the next few years (prior to 2020) where sufficient published and unpublished data were available (Table 2). For simplicity insecticides containing the appropriate concentration of pirimiphos methyl and clothianidin are subsequently referred to by their product names Actellic®300CS and SumiShield®50WG, respectively. The impact of IRS depends on its initial efficacy and how this changes

over time. Studies with 3 or more experimental hut trial time-points were considered sufficient to characterise temporal changes. Reasons for excluding studies are noted in Supplementary Table 1.

Altogether 8 published and 1 unpublished studies (providing 21 time series) were found that reported experimental hut trials on pyrethroid-IRS[17,25,63–68] (Table 2). Three published studies and a further unpublished dataset were identified for bendiocarb[39,63,66]. Four published and 1 unpublished studies provided 6 time series data for Actellic®300CS[25,26,63,65]. In total, 2 published and 2 unpublished datasets were available for SumiShield®50WG[67,68]. This new formulation was tested at different concentrations and we include concentrations of 300 g m$^{-2}$ and above in the presented analyses.

**Delayed mortality**. The mode of action of the neonicotinoid insecticide clothianidin has been shown to act over multiple days on the insect's nervous system so the 24-h mosquito mortality measured in a SumiShield®50WG experimental hut trial is unlikely to fully represent the efficacy of this chemistry[69]. To generate comparable parameterisation of products with different modes of action a simple conversion is used to convert 72-h experimental hut trial mortality rates into 24-h mortality rates that can be used in the transmission dynamics model. This is possible if it is assumed that SumiShield®50WG exposed mosquitoes have no epidemiological impact between 24 and 72-h following exposure, which, given the frequency of blood-feeding, appears the most parsimonious assumption. If mosquitoes caught in the other arms of the trial (untreated huts and those with fast acting chemistries) die at a constant rate between 24 and 72-h, then the background mosquito death rate for a mosquito in captivity following a hut trial can be estimated using the exponential function. If $l_{SC}(t)$ denotes the proportion of mosquitoes that are dead in captivity $t$ days after the start of the hut trial, then the background mortality rate ($\mu_B$) can be estimated using,

$$l_{SC(t)} = 1 - \exp(-\mu_B \times t). \quad (6)$$

Fitting this function to all datasets where 72-h ($t = 3$ days) mortality were recorded gave $\mu_B = 0.035$. This value was then used to adjust the mortality observed 72-h after the start of the SumiShield®50WG trials to generate estimates of 24-h mortality comparable to the other insecticides.

**Successful blood-feeding**. Data were not always disaggregated by the mosquitoes that had fed and survived or fed and died. Therefore, it was not possible to directly infer which mosquitoes were successfully feeding. Instead, before fitting the time series data, we adjusted the number of mosquitoes that were blood feeding ($N_{fed}$) to provide an estimate for the successful blood-feeding mosquitoes ($N_{successfully\_fed}$), those that feed and survive, as follows:

$$N_{successfully\_fed} = N_{fed} \times \left(1 - \frac{N_{dead}}{N_{total}}\right) \quad (7)$$

$N_{total}$ and $N_{dead}$ denote the total number of mosquitoes sampled and the total number recorded as dead for each time series.

**Model fitting**. Logistic binomial models were fitted to the count data to determine the relationship between the probable outcome of a mosquito feeding attempt (the mosquito is deterred, killed, successfully feeds or exits without feeding) and how these change over time. Briefly, to determine, for example, the relationship for the proportion of mosquitoes that are killed ($l_s$) in the presence of an IRS product over time $t$, we fit:

$$l_S = logit^{-1}(p) = \frac{1}{\left(1 + \exp^{-(l_{S\beta} + l_{S\gamma} \times t)}\right)} \quad (8)$$

$$N_{dead} \sim binomial(logit^{-1}(p), N_{total}) \quad (9)$$

The proportion of mosquitoes dying following entering a hut is denoted $l_S$ and is dependent on a parameter that determines initial efficacy ($l_{S\beta}$) and how this changes over time, denoted by the depreciation parameter ($l_{S\gamma}$). The mosquitoes that are successfully feeding or deterred are modelled in the same way (Supplementary Methods). These different probable outcomes of a feeding attempt are then translated into the probability of a mosquito being killed, successfully feeding or being repelled as detailed in Supplementary Methods. To determine uncertainty, the maximum and minimum data for each unique time point were fitted in the same way. The ranges for the probability of mosquitoes successfully feeding, exiting or being killed at each feeding attempt in the presence of each IRS product could then be distinguished (Supplementary Fig. 8). As previously, Bayesian models were fitted using Hamiltonian Monte Carlo sampling methods[60,61]. The fits were visually confirmed to fit the data (Fig. 2).

**Transmission dynamics mathematical model**. A widely used transmission dynamics model of malaria[2,70–72] is used to investigate the public health impact of different IRS compounds. In this model, people are born susceptible to *Plasmodium falciparum* infection and are exposed to infectious mosquito bites at a rate

dependent on local mosquito density and infectivity. Maternal immunity is acquired for new born infants and this decays in the initial 6 months of life. Individuals are susceptible to clinical and severe disease and death after exposure[71,72]. The risk of developing infection declines with age due to naturally acquired immunity following continual exposure. Mosquito dynamics capture the effects of mosquito control and the resulting decline in egg laying[70]. A small number of minor changes (see Supplementary Methods) are adopted to the IRS component of the model to reflect the varying impact of the new chemistries and how these change over time. These changes unify the way LLINs and IRS are represented in the model (and are parameterised with experimental hut trials) and provide greater flexibility to capture the impact of different insecticides. The transmission model is used to simulate across the parameter ranges (Supplementary Table 3) to explore the minimal and maximal entomological impact of a given product and the knock-on predicted impact on cases (Supplementary Data 2).

**Pyrethroid resistance**. Discriminating dose bioassays (WHO tube assay, WHO cone assay, CDC bottle assay) are a practical option for control programmes to assess the proportion of the mosquito population that are killed by a standard dose. The assumption is made that the inverse of this proportion, i.e. those mosquitoes surviving in the presence of the standard dose of insecticide, is representative of the level of insecticide resistance in the mosquito population. Although the simple bioassay has its limitations[20,37] it provides a useful measure to link the severity of mosquito insecticide resistance estimated in the field to the results of experimental hut trials evaluating new products[20,34]. The concentration of insecticide used in the discriminatory dose bioassay varies with the type of pyrethroid insecticides used. There were 18 data points identified in the meta-analysis where pyrethroid bioassays were conducted on the same mosquito population as the experimental hut studies using a pyrethroid IRS (Fig. 3a). There were a further 21 datasets with time series data, but not bioassay mortality data, so that the initial (time $t = 1$ day) mosquito mortality, successful feeding and exiting probabilities and how the impact of pyrethroid IRS on mosquito behaviour changes over time could be estimated. This is insufficient data to differentiate between different types of pyrethroid so all pyrethroid data are pooled together, recent work suggests this may be a reasonable assumption[73]. These two datasets are used to associate 24-h mortality using a discriminatory dose bioassay (our proxy for the level of pyrethroid resistance in the mosquito population) and the parameters influencing IRS efficacy (see Supplementary Methods). These changes are demonstrated in Supplementary Fig. 9. The code for analyses 1–3 are provided in Supplementary Methods.

**Utility of the model**. RCT are the gold standard for assessing intervention efficacy and effectiveness in the field. Results from two RCTs testing the additional benefit of Actellic®300CS[8] or bendiocarb[38] IRS in combination with standard LLINs over standard LLINs alone were compared to model predictions to determine whether the IRS parameterisations satisfactorily match observed data. The location-specific seasonality profile and historic bednet use were taken from[1,40,74], whilst study-specific parameters such as epidemiological information (for example cohort age), intervention type (for example decay of net use) and mosquito information (for example the ratio of different mosquito species present and the level of pyrethroid resistance) were taken from the relevant publications and discussions with study authors (see Table 3 for a summary of input parameters). Predictions were made for all RCT data combined without differentiating between clusters. Absolute mosquito abundance is varied to ensure model predictions at baseline match the average malaria prevalence for the age cohort examined. Future predictions are then made using the model and compared visually to observed RCT prevalence measured during cross-sectional surveys. The parameter sets fitted to the temporal data (Supplementary Data 1, analysis 2) that describe the mean impact, as well as the maximum and minimum impact (Supplementary Methods for parameter estimates), of the respective IRS products on mosquito feeding outcomes were used to predict the public health impact. The parameter sets for each individual study (Supplementary Fig. 6) were also overlaid to provide some indication of the potential uncertainty in the experimental hut data. For the Actellic®300CS trial[8], the best-matched experimental hut data were from Rowland et al.[25] which took place in Benin and used West African huts with both cement and mud walls. The principal mosquito in both localities (the Benin experimental hut trial and the Kagera RCT) was *A. gambiae s.s.* and houses in Kagera also have, most commonly, cement or mud walls.

**Model simulations**. IRS and LLINs are used concurrently (i.e. the same people receive IRS and LLIN) in many malaria endemic communities[75]. The efficacy of IRS on top of LLINs will depend on LLIN coverage, the level of pyrethroid resistance and the seasonality of malaria transmission. To illustrate how these factors influence disease control, the transmission model is parameterised for a theoretical perennial and a highly seasonal setting. For simplicity all simulations are initially run in an area with moderate transmission (slide prevalence of 30% without intervention or treatment) in an area with no history of malaria control. At the start of year 0, LLINs are distributed at a pre-determined coverage level (the percentage of people who use them) ranging from 0 to 100%. The net usage remains at this level for the whole simulation. Pyrethroid resistance is simulated to arrive overnight at a defined level (as described by the percentage of mosquitoes

**Table 3 Site-specific factors used to parameterise the transmission dynamics model**

| Study | West[38] | Protopopoff[8] |
|---|---|---|
| Year | 2011–2012 | 2015–2017 |
| Malaria prevalence diagnostic | Rapid diagnostic tests | Rapid diagnostic tests |
| Cohort survey age range | 0.5–14 years | 0.5–15 years |
| Baseline malaria prevalence | 24% LLIN only arm | 68% |
|  | 21% LLIN+IRS arm |  |
| Percentage of mosquitoes *A. gambiae s.s.*[a] | 80% | 91.6% |
| Percentage of mosquitoes *A. arabiensis*[a] | 20% | 4.6% |
| Percentage of mosquitoes *A. funestus*[a] | 0 | 3.8% |
| Bioassay mortality (*A. gambiae s.l.*)[a,b] | 11%[76] | 7.8% |
| Bioassay mortality (*A. funestus s.l.*)[a] | – | 54.5% |
| IRS insecticide | Bendiocarb | Pirimiphos methyl |
| IRS coverage (%) | 5% (LLIN only arm) | 94% |
|  | 90% (LLIN+IRS arm) |  |
| IRS timing | December 2011 and May 2012 | February 2015 |
| LLIN coverage (%) | 53% LLIN arm | 78% |
|  | 58% LLIN+IRS arm |  |
| Average duration of LLIN use (bednet usage assumed to decline at a constant rate) | 3.636 years | 2.813 years |

Site-specific factors used to parameterise the transmission dynamics model and investigate its ability to predict the impact of IRS in the two randomised control trial (RCT). The efficacy of standard LLINs and pyrethroid IRS is adjusted according to the mosquito mortality in the discriminating dose bioassay using the adjustments noted in Supplementary Methods and methods presented in Churcher et al.[20] for standard LLINs. All other parameters are consistent with Griffin et al.[2], White et al.[70], Griffin et al.[71], and Winskill et al.[40] (for seasonality profiles, historical intervention coverage, drug treatment information). There is insufficient data to characterise whether the mosquito species distribution or the level of pyrethroid resistance changed over time, so these are assumed to have remained constant throughout in all intervention arms. Systematic non-compliance is assumed in arms where both LLINs and IRS were distributed
[a]Denotes parameters which are kept constant throughout the simulations for all intervention arms
[b]The same parameter estimates are used for *A. arabiensis*

surviving a 24-h discriminating dose bioassay test, 0–100%) with the introduction of nets at year 0. At the start of year 3, IRS is introduced, be it a long-lasting product (e.g. Actellic® or SumiShield®), a short-lasting product (e.g. bendiocarb) or a pyrethroid performing at the defined level of resistance. SumiShield® produced broadly similar results to Actellic® and is not represented in Fig. 5. Eighty percent of the people are protected by the IRS. The households are sprayed just prior to the peak in the transmission season each year. A 3-yearly reporting cycle is adopted to coincide with the redistribution of LLINs (generally every 3 years). The clinical cases averted per 1000 people per year by the respective IRS chemistries are calculated between years 3 and 6 relative to a scenario for the same level of pyrethroid resistance where no IRS is implemented.

Every area of Africa has a different malaria seasonality and history of control interventions so the impact of adding different types of IRS will vary locally. To facilitate assessment of the public health and economic benefit of adding different IRS options their impact is simulated at each administration unit 1 across sub-Saharan Africa and at increasing levels of pyrethroid resistance. Location-specific seasonal profiles[74] and historic scale-up of IRS and LLIN interventions from 2000 to 2015 are used (Malaria Atlas Project, MAP[1]) as per[40]. The mosquito density is adjusted for each location to capture the underlying transmission intensity and ensure model predictions match MAP estimates for *P. falciparum* prevalence in 2–10-year olds. Mosquito densities are then scaled up for each country so that the total cases estimated is equal to the WHO estimate for 2015 for that country whilst also capturing administration-level heterogeneity in transmission[40]. Pyrethroid resistance is switched on overnight in 2018, whilst maintaining 2015 net coverage levels. A 3-yearly reporting cycle is once again adopted to coincide with the redistribution of LLINs (generally every 3 years). IRS is introduced at 80% coverage in 2021 using either long-lasting (Actellic® or SumiShield®), short-lasting (bendicarb, with annual or biannual application) or pyrethroid-IRS for distinct levels of pyrethroid resistance (ranging from 0 to 100%) (Supplementary Data 2). The IRS product parameterisations are the mean fits for the temporal data (Supplementary Table 3), with the exception of pyrethroid IRS, which is complicated by the presence of resistance in local mosquito populations. To provide some indication of the uncertainty in these product impacts, the transmission model is also used to simulate the predicted maximum and minimum impact of non-pyrethroid IRS products as estimated from the temporal analysis (Supplementary Data 1, analysis 2). Bendiocarb was modelled annually and biannually because it is usually sprayed twice a year if used for IRS programmes. The mean number of cases averted per 1000 people per year across the following 3 years, 2021–2024, was then calculated relative to a scenario where no IRS was used.

### Data availability

The authors declare that all published data collated during the systematic review supporting the findings of this study are available within the paper and its supplementary information files. The unpublished data that support the findings of this study are available from the corresponding author upon reasonable request and in agreement with the data owners. The transmission model parameters that are used to define specific administration units across Africa can be provided upon request.

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

## Acknowledgements

The work was supported by the Innovative Vector Control Consortium (IVCC), the Wellcome Trust [200222/Z/15/Z] MiRA and the UK Medical Research Council (MRC)/UK Department for International Development (DFID) under the MRC/DFID Concordat agreement. Thank you to all who conducted the experimental hut trials.

## Author contributions

T.S.C., J.H.R. and E.S.-S. designed the project. V.C., C.P., A.D., S.M., P.M., C.E., N.P., R.O., F.A., R.N. and M.R. collected data, performed experimental hut trials and randomised control trials. E.S.-S., T.S.C., J.T.G. and P.W. analysed the data. E.S.-S. collated the data and produced all the figures. T.S.C. supervised all aspects of the project. E.S.-S. and T.S.C. wrote the paper. All authors discussed results, commented on and approved the final manuscript.

## Additional information

**Competing interests:** The authors declare no competing interests.

