## [Peer Review File · Nature Communications]

Reviewers' comments:

Reviewer #1 (Remarks to the Author):

This work will be a substantial contribution to the community, following a similar meta-analysis of ITN effects (eLife 2016).

However, the way that site-specific variation is handled in the estimation of time-dependent mortality is problematic, especially for pirimiphos-methyl.

The downstream results -- model comparisons to Tanzanian prevalence measurements and Africa-wide projections -- suffer from an overly simplistic treatment of this and other significant sources of uncertainty.

More work needs to go into the treatment and propagation of uncertainty if the latter sections are to be included in this paper. If this can be done, I would look forward to reviewing an updated draft.

=====

L160, L181, L379-284: There is substantial variation among sites, which is likely related to species and hut design, but the limited data don't allow a definitive allocation to specific covariates. Ignoring that information and fitting an average is not an adequate approach, especially for comparisons to specific out-of-sample sites or Africa-wide projections.

Fig 1:

- first-column label typo
- a version with numbered, site-specific fits would tell a slightly different and complementary story.
- move symbol legend to main figure instead of burying in caption (e.g. "cross" is a bit of an ambiguous symbol name: x or +)
- Bendiocarb deterrence seems to be a very poor fit (L172)?
- The extremely different patterns of Clothianidin deterrence after 4 months is worth a comment somewhere. I'm not sure I quite understand how the Benin deterrence and successful blood-feeding can be consistent? Perhaps it relates to the definitions of these quantities in the Methods section?

Fig 2:

- L219: "some mosquito mortality due to IRS is predicted even when all mosquitoes are surviving the discriminatory dose bioassay"
- Does the fit in Fig. 2a consider the binomial errors on mosquito-survival counts? Total counts vary from 29 to 1149, so this is not a trivial consideration.
- From a quick glance at the "Pyrethroid Resistance Data" table in Supplementary Data 1, it would seem that the point with 0% survival on the pyrethroid resistance test is from the excluded unpublished data.
- Should the fit be constrained to go through (0, 0) from first principles?
- Could horizontal and vertical error bars be added to points as appropriate?
- Could points in Fig. 2d be numbered (or at least use consistent markers) by site for comparison with Figure 1?

Fig 3:

- These are interesting illustrations of the utility of the fitted IRS (and ITN) parameters in an out-of-sample use case, but I don't think they add much direct "validation" of the hut trial fits.
- If you intend to frame this section as "validation" of hut-trial fits, a deeper inspection of the Tanzanian trial data would be required: e.g. personal protection vs. community effect, longitudinal

reinfection of individuals, heterogeneity in exposure, seasonal and year-on-year variation. The cluster-level information alluded to in L558 might be informative?

- Would an Actellic decay profile other than the average fit -- but still consistent with one of the longer-lasting sites -- give a more realistic fit to the 2016 measurement points? This would seem like a useful place to compare how the site-specific variation -- site 7, site 4 (mud), site 4 (cement) -- manifest themselves in a real-world example.
- Is there any information about the house construction in Kagera that would change the prior expectation of Actellic decay (as discussed in L384-387)?

Fig 4:

- L341: The "drive for malaria elimination" is an important consideration.
- Is there a supplemental figure in the style of Fig. 4 that could be used to illustrate that more clearly, e.g. the scale reduction in vectorial capacity as opposed to the moderate-endemicity cases averted?

Fig 5:

- The interpretation of these maps are somewhat problematic, since they are very likely to give a false sense of precision.
- The reader will not necessarily have an intuitive sense of what transformations of impact would occur if one were able to convolve with the effective pyrethroid resistance map.
- The reader is not presented with information to assess the veracity of mosquito indoor-resting assumptions by Admin-1 and how that would relate to IRS/ITN impact.
- The reader has the tabular ITN coverage assumptions in Supplemental Data 2, but could easily be presented with this information in a map.

L397-401: This is an interesting point, that I would strongly agree with. I do wonder, though, given the species- and structure-dependent variation in effectiveness, whether non-inferiority on a given surface in a given site should be considered sufficient evidence, or whether the authors have some thoughts on what range of settings would be sufficient?

Reviewer #2 (Remarks to the Author):

Summary

This an interesting study of a very important topic. The first part of the paper describes a highly informative set of meta-analyses and the modelled effects of the associations found on the efficacy of indoor residual spraying, an important tool in the drive to control malaria. The second part of the paper goes on to describe a series of malaria transmission model simulations, however, this latter part of the study lacks validation. The data available to parameterise and validate the transmission model is a substantial limiting factor.

Major comments

The authors provide four formulae describing how to work out a percent but then skimp on the details of the actual meta-analyses conducted and the model used for the simulation. The methods for both need to be clearly summarised for a Nature Communications reader in the main methods, with full details provided in the supplement to allow others to repeat the analyses. Stating that effects are "quantified statistically" without providing any more information in the main manuscript is inadequate.

Figures 1 and 2 have shaded areas around the lines but there is no indication of what the shaded areas represent, or how the lines were generated (see the point above).

Only the modelled additional benefits of pirimiphos methyl and bendiocarb in combination with

permethrin-treated nets were validated, and only under a limited set of conditions (a single district in Tanzania sampled during two time periods). The model is then used to predict a much wider range of effects in a much wider range of settings but none of these predictions are validated. This is a major limitation of this work and needs to be clear in statements relating to the simulation results. No indication of uncertainty is provided for any of the predictions given in the results (lines 277-303 and figures 4-5).

Minor comments

Lines 49-53. Ref. 1 showed that the coverage of ITNs and IRS that was achieved in Africa during 2000-2015 had a larger impact than the coverage of ACTs that had been achieved during the same period, that is not the same as showing that ITN/IRS are more effective tools than ACT. The results of refs 2-3 are stated as fact but they are actually predictions.

I recommend getting someone to proof read the Introduction to ensure it flows well and that the use of parentheses within parentheses doesn't render sentences hard to read. For example, within lines 60-77 the sentences don't always follow on from each other in an obvious way and the final sentence seems like an outlier. The Discussion is much better.

The search for studies that contributed to the meta-analysis was conducted on 25 May 2015, which is over three years ago. Given the lack of data, it's surprising the authors didn't update this search.

Lines 462-465. As Nature Communications has a broad authorship I recommend defining 'exophily' (not just the mathematical/experimental hut definition). It may also be worth including a brief explanation of how an experimental hut trial works.

Lines 528-533. Be careful about stating WHO tube assays, etc assess the 'level' or 'severity' of resistance. Unless concentration is varied (which is rarely the case) they only measure the proportion of mosquitos from a population sample killed by a standard dose.

Lines 531-541. A recent study (Hancock 2018 PNAS 115:5938) provides further justification for pooling pyrethroid data although this is still a limitation of the study as some differences are found.

Lines 548-561. The species composition in Table 4 doesn't match that reported by Protopopoff but it's not clear why. What 'level' of resistance was assumed for funestus?

Lines 383-4. There is evidence to support the statement about resistance varying between species, e.g. Ondeto (2017) P&V 10:429, etc, etc.

Lines 385-387. Were data from West African hut designs excluded from all parts of the study including the information that went into the simulations across the whole of Africa? This really highlights the limitation of validating part of the work only, using data from one (eastern) location and then simulating transmission across many scenarios. I didn't spot any mention of this exclusion in the Methods or the Results, just the Discussion.

It needs to be clear when the results are presented which come from a model that assumes moderate (30%) endemicity and no prior interventions, and which come from a model that uses data for seasonal profiles and data on past scale-up of IRS and LLIN.

Check the meaning of principle vs principal.

Reviewer #3 (Remarks to the Author):

NCOMMS-18-18452

The authors have conducted a meta-analysis of experimental huts trials to characterize the efficacy of different IRS products on anophelines. These data have been used to parametrize an established model and predict the impact of different IRS product classes across a range of eco-epidemiological situations with varying LLIN coverage and pyrethroid resistance. The results are compared against two published trial for validation with relative success.

This allows for efficacy and cost assumptions in different scenarios which can be an asset for NMCP managers and those involved in drafting national strategic plans. Donors and implementers could also benefit. Assumptions are well laid out and limitations are reasonably explained. The conclusions are supported by the results.

There are however some methodological issues that require clarification.

MAJOR

The authors refer to the background review as a “systematic meta-analysis” (line 440). PRISMA guidelines are mentioned in line 439.

The PRISMA checklist (<http://prisma-statement.org/prismastatement/Checklist.aspx>) however has not been followed. Some examples are:

- The title does not identify the report as a systematic review or meta-analysis (or both)
- There is no reference to a protocol or registered methodology
- The description of search methods and assessment of title/abstracts is limited

MINOR

Introduction:

There is no mention of residual transmission as a challenge although the mosquito behavior leading to it is referred to in lines 373-375

Authors should mention and include in the references the WHO guidance for countries on combining indoor residual spraying and long-lasting insecticidal nets.

Lines 116-119: should mention and include in references the WHO information note (2017) on the evaluation process for vector control products.

Lines 76-77: the phrase is incomplete “LLINs still provide personal protection by acting as a physical barrier and invoking minimal killing and some repellence”. In presence of resistance I presume?

Results:

Lines 161-162: please elaborate further in the discussion on the potential implications of combining the results of *An. gambiae* s.l. and *An. funestus* as well as different hut designs and substrates.

Discussion:

Lines 420: 80% coverage might not be financially or operationally achievable everywhere or with all chemistries. Different chemistries have different effect on the walls (staining) and different smells which directly affects acceptability.

Itemised response to reviewers' comments:

Reviewer #1 (Remarks to the Author):

This work will be a substantial contribution to the community, following a similar meta-analysis of ITN effects (eLife 2016).

However, the way that site-specific variation is handled in the estimation of time-dependent mortality is problematic, especially for pirimiphos-methyl.

The downstream results -- model comparisons to Tanzanian prevalence measurements and Africa-wide projections -- suffer from an overly simplistic treatment of this and other significant sources of uncertainty.

More work needs to go into the treatment and propagation of uncertainty if the latter sections are to be included in this paper. If this can be done, I would look forward to reviewing an updated draft.

We agree, we have redone the meta-analysis to update this to 27th July 2018. We have fitted a logistic binomial model to these data and tested the impact of mosquito species, hut type, wall substrate and IRS product on the immediate efficacy (impact within 2 months since spraying) to better appreciate some of the causes of uncertainty in IRS performance (new Fig. 1). Using the additional data sources identified, we have re-fitted the functions capturing probable outcomes of mosquito feeding attempts to experimental hut data. To try to capture the uncertainty better, we have taken the parameter estimates describing the best and worst performing studies for each IRS product and then modelled RCTs by drawing from this range of estimates for the parameters. This generates a much bigger uncertainty (new Fig. 4). To carry this forward, we produce maximum and minimum estimates, using these broad parameter estimates, for cases averted to complement the table of mean cases averted in Supplementary Data 2. We can also provide the relevant parameter estimates for the specific sites that were used as default estimates for specific administration units if requested.

=====

L160, L181, L379-284: There is substantial variation among sites, which is likely related to species and hut design, but the limited data don't allow a definitive allocation to specific covariates. Ignoring that information and fitting an average is not an adequate approach, especially for comparisons to specific out-of-sample sites or Africa-wide projections.

Agreed, we now present a statistical analysis of the initial data to show differences between IRS products, mosquito species, wall substrates and hut types (restricting this to East or West African hut design) (new Fig. 1 and see Supplementary Figures 2 – 5). The limited data for time series analyses do not allow us to understand how these covariates might alter the duration of impact. Instead, to present the uncertainty, we now used the best and worst performing experimental hut studies for each IRS product and then generate estimates that span this range, rather than presenting a single average estimate. We add a minimum and maximum tab for each IRS product (Fig. 2, Supplementary Figure 9) to demonstrate the effect of this range on IRS predicted performance. We also make it clearer that there is substantial uncertainty and limitations to the analysis within the text [lines 167 - 182, 462 – 479 and elsewhere].

For the location-specific predictions, we use site-specific parameters to capture the seasonality, historic net and IRS use and mosquito species compositions within administration sites. We can provide these for each administration unit if useful.

Fig 1: (new Fig. 2)

- first-column label typo

Corrected

- a version with numbered, site-specific fits would tell a slightly different and complementary story.

Added as Supplementary Figure 6.

- move symbol legend to main figure instead of burying in caption (e.g. "cross" is a bit of an ambiguous symbol name: x or +)

Done, the symbols now relate to data resources collated in Table 2. (We use these symbols for Fig. 2, 3 and Supplementary Figure 6).

- Bendiocarb deterrence seems to be a very poor fit (L172)?

We have had challenges with how to assess deterrence because experimental huts on IRS are not able to rotate products very easily. This could mean that mosquitoes feeding principally on sprayed huts are killed off so that the total number visiting these huts decreases. In the data this presents as increasing deterrence as this is essentially the ratio between control and sprayed huts. We have now made the biologically plausible assumption that the impact of the IRS in deterring mosquitoes wears away at a rate matching the depreciation of the mortality effect. We have fitted to the initial deterrence and substitute in this depreciating parameter (Fig. 2 – row 3). Parameter estimates are provided in Supplementary Table 3.

- The extremely different patterns of Clothianidin deterrence after 4 months is worth a comment somewhere. I'm not sure I quite understand how the Benin deterrence and successful blood-feeding can be consistent? Perhaps it relates to the definitions of these quantities in the Methods section?

In our approach, the probability plots in row 4 represent the total mosquito population but only the mosquitoes that are not deterred, and enter huts, can then either successfully feed, exit or are killed. As noted above, there is a challenge in measuring deterrence over time because we cannot rotate IRS products between sites. We consider the deterrence data showing increasing deterrence as an artefact given there is no sensible biological explanation for this pattern. We assume the same depreciation of deterrence efficacy as mortality efficacy. However, we do comment on this challenge within the text [lines 473 - 475, Fig. 2 legend and see Supplementary Methods 2].

Fig 2: (new Fig. 3)

- L219: "some mosquito mortality due to IRS is predicted even when all mosquitoes are surviving the discriminatory dose bioassay"

- Does the fit in Fig. 2a consider the binomial errors on mosquito-survival counts? Total counts vary from 29 to 1149, so this is not a trivial consideration.

Yes, the fit is weighted by the total counts of mosquitoes in each study. To clarify, we have added the binomial distributions to the descriptions of the statistical model fitted to the initial data (and also for the time-dependent fits for all IRS products) to the methods (and Supplementary Methods 2).

- From a quick glance at the "Pyrethroid Resistance Data" table in Supplementary Data 1, it would seem that the point with 0% survival on the pyrethroid resistance test is from the excluded unpublished data.

To clarify, we have now altered the symbols in new Fig. 3 to correspond to the data presented in Table 2. We can see that the 0% bioassay survival is taken from the unpublished data (reference 11 in Table 2).

- Should the fit be constrained to go through (0, 0) from first principles?

We think it is possible that free-flying mosquitoes have some probability of dying even where all mosquitoes are killed by a standard dose WHO bioassay and so have opted not to constrain the prediction to 0. Free-flying mosquitoes may be old, previously damaged or killed by someone who is awake and defending themselves so in the absence of intervention this estimate might not need to be restricted.

- Could horizontal and vertical error bars be added to points as appropriate?

We have used point estimates for these data.

- Could points in Fig. 2d be numbered (or at least use consistent markers) by site for comparison with Figure 1?

Thank you, yes we have made symbols consistent with figure 1 (now Fig. 2) (and also Supplementary figure 6 which shows the individual fits for time series data).

Fig 3: (new Fig. 4)

- These are interesting illustrations of the utility of the fitted IRS (and ITN) parameters in an out-of-sample use case, but I don't think they add much direct "validation" of the hut trial fits.

- If you intend to frame this section as "validation" of hut-trial fits, a deeper inspection of the Tanzanian trial data would be required: e.g. personal protection vs. community effect, longitudinal reinfection of individuals, heterogeneity in exposure, seasonal and year-on-year variation. The cluster-level information alluded to in L558 might be informative?

We have opted to adjust the wording of this section, have improved the parameterisation of the overall predictions and have included the variation in the predicted change in prevalence using the maximum and minimum performance of pirimiphos-methyl products. We use the seasonal profile for the administration unit of Kagera, Tanzania rather than re-fit to rainfall for the respective years of these RCTs. We now use these figures to illustrate the utility of the IRS fitting rather than considering these as validation.

An assessment of the cluster-level information would be of interest. If we were to proceed with fitting the individual clusters, we would also choose to fit rainfall data for the different clusters. This process would take some time and, given that the current manuscript is already substantial, we consider this to be a distinct piece of follow-on work were the data owners happy to share the data.

- Would an Actellic decay profile other than the average fit -- but still consistent with one of the longer-lasting sites -- give a more realistic fit to the 2016 measurement points? This would seem like a useful place to compare how the site-specific variation -- site 7, site 4 (mud), site 4 (cement) -- manifest themselves in a real-world example.

This is a really good idea thank you. We now show this in the figure by adding distinct dashed lines to indicate the predictions from the different Actellic or Bendiocarb studies. We overlay these onto the polygons showing the maximum and minimum range predicted and the uncertainty of other model parameters around the mean parameter fit for the IRS product in question. In this way, we can comment on the different experimental hut outcomes and the uncertainty in the potential impact of IRS in site-specific locations.

- Is there any information about the house construction in Kagera that would change the prior expectation of Actellic decay (as discussed in L384-387)?

The Muleba district in Kagera has houses made with different substrates (mud, brick, cement, plastered lime)¹. In the recent RCT testing Actellic®300CS² the mosquitoes were principally *An. gambiae s.s.* (91.6%) and most of these mosquitoes could survive in the presence of the standard dose of deltamethrin tested in WHO bioassays (7.8% 24-hour mortality). The experimental hut data on Actellic®300CS was mostly on *An. gambiae s.l.* and tested in cement-walled West African huts³⁻⁶. Rowland et al.⁴ also provided data for mud-walled West African experimental huts which performed a little worse than the IRS on cement (see new Fig. 4). IRS products performed less well on mud substrate in the initial impact analysis (new Fig. 1). We have commented on this in the text [lines 167 – 182, 307-318].

Fig 4: (new Fig. 5)

- L341: The "drive for malaria elimination" is an important consideration.

- Is there a supplemental figure in the style of Fig. 4 that could be used to illustrate that more clearly, e.g. the scale reduction in vectorial capacity as opposed to the moderate-endemicity cases averted?

This is a good idea but given the already substantial length of the submission, we would prefer to consider exploring this within a separate piece of work. We are considering the general effect of IRS and the situation is more nuanced as a setting tends toward elimination.

Fig 5:

- The interpretation of these maps are somewhat problematic, since they are very likely to give a false sense of precision.

- The reader will not necessarily have an intuitive sense of what transformations of impact would occur if one were able to convolve with the effective pyrethroid resistance map.

- The reader is not presented with information to assess the veracity of mosquito indoor-resting assumptions by Admin-1 and how that would relate to IRS/ITN impact.

- The reader has the tabular ITN coverage assumptions in Supplemental Data 2, but could easily be presented with this information in a map.

Thank you, we agree that these could be misleading and have opted to remove this figure altogether. Instead, we have added in, to Supplementary data 2, a minimum and maximum estimate for cases averted given the range in parameter estimates for the data from new Fig. 2 and Supplementary Figure 6. We have then expanded on the explanation of these supplements to try to demonstrate the uncertainty more clearly and avoid mis-interpretation.

L397-401: This is an interesting point, that I would strongly agree with. I do wonder, though, given the species- and structure-dependent variation in effectiveness, whether non-inferiority on a given

surface in a given site should be considered sufficient evidence, or whether the authors have some thoughts on what range of settings would be sufficient?

Given that the majority of the data we have explored tests *An. gambiae ss* on either cement or, more rarely, mud substrate, and we have relatively consistent predictions for the West African experimental hut data within these categories, it seems reasonable to argue non-inferiority within these tests could be sufficient. However, there is an unknown impact on alternative mosquito species and on alternative wall substrate. We do not consider social differences in activity or net use here. These will also have arguably substantial impact on next-generation net and IRS performance. In communities that are active throughout the night or sleep outdoors or have particular social systems that differ to those in a tested scenario, the proportion of bites that are received either indoors or in bed will be different and will alter the optimal impact that indoor interventions can achieve. There is a concerning train of thought that IRS achieving less than 80% mortality is insufficient. The assessment of pyrethroid resistance clearly indicates that sub-optimal effects are still better than not using a spray. This is a critical message but perhaps for a future piece of work.

Thank you for the very constructive criticism.

Reviewer #2 (Remarks to the Author):

Summary

This is an interesting study of a very important topic. The first part of the paper describes a highly informative set of meta-analyses and the modelled effects of the associations found on the efficacy of indoor residual spraying, an important tool in the drive to control malaria. The second part of the paper goes on to describe a series of malaria transmission model simulations, however, this latter part of the study lacks validation. The data available to parameterise and validate the transmission model is a substantial limiting factor.

Major comments

The authors provide four formulae describing how to work out a percent but then skimp on the details of the actual meta-analyses conducted and the model used for the simulation. The methods for both need to be clearly summarised for a Nature Communications reader in the main methods, with full details provided in the supplement to allow others to repeat the analyses. Stating that effects are “quantified statistically” without providing any more information in the main manuscript is inadequate.

We agree, we have now updated the meta-analysis and provide a systematic summary of the process in Supplementary methods 1. The summary review is provided in Supplementary Fig. 1 and our reasons for rejecting data resources are noted in Supplementary Table 1. Here, we highlight the studies used in each analysis:

1. The summary data analysis of the initial impact (impact within 2 months since spraying an IRS compound) follows most closely to PRISMA guidelines. We have now added the binomial logistic regression that is used for these assessments and present the data in new Fig. 1 which contrasts different products. Supplementary Table 2 provides further detail in these contrasts using a Bayesian approach. We expand on this in the main manuscript [lines 154 – 196 (results), 559 – 585 (methods)]. (Supplementary Data 1, analysis 1)
2. The data that can be used to assess the functional impact of IRS compounds over time since spraying. Here, we fit binomial probability distributions to the mortality, successful blood-feeding (fed and survived) and deterred data. We explain that, given the uncertainty in measuring deterrence for IRS studies, we assume the same decay rates for the loss in deterrence for each product as that which can be measured using the mortality data. To capture the variation between studies and hence the uncertainty in this approach, we now parameterise the best and worst performing studies for each IRS product and then complete a sensitivity analysis across the range of parameters defining these levels of performance. (Supplementary Data 1, analysis 2)
3. Our third analysis focuses on pyrethroid resistance. Here we look to predict the lost impact of pyrethroid IRS as mosquitoes are able to survive in the presence of a standard dose of pyrethroid as measured in a WHO tube bioassay. We make the assumption that this is representative of the level of pyrethroid resistance in a mosquito population. (Supplementary Data 1, analysis 3a and b)

We now use these estimates of uncertainty in the transmission model to explore how well the model can predict randomised control trials. We correct the data for the RCT in Tanzania which, as you noted, was a mis-match for the information in the paper – apologies for this error. We have

made our assumptions clearer by presenting additional parameter information in Supplementary data 2 and have outlined limitations more clearly in the discussion.

Figures 1 and 2 have shaded areas around the lines but there is no indication of what the shaded areas represent, or how the lines were generated (see the point above).

We have added these analyses to the methods and add further detail in Supplementary Methods 2. However, we have updated Figures 1 and 2 (new Fig. 2 and 3) based on suggestions from reviewer 1. We now show the fits as previously but also show the range in parameter predictions (pale-shaded polygon) and the weighted-mean fits to the combined data including each time series as a random effect. We have updated the legends and keys to better explain these figures. The darker-shaded areas are the 90% credible intervals around the mean fit for the change in mosquito mortality, successful feeding or deterrence over time. We also include an additional figure (Supplementary Figure 6) to demonstrate the study specific data which we hope complements new Fig. 2. The stan code for the statistical models included in each analysis are now provided as Supplementary Methods 3.

Only the modelled additional benefits of pirimiphos methyl and bendiocarb in combination with permethrin-treated nets were validated, and only under a limited set of conditions (a single district in Tanzania sampled during two time periods). The model is then used to predict a much wider range of effects in a much wider range of settings but none of these predictions are validated. This is a major limitation of this work and needs to be clear in statements relating to the simulation results. No indication of uncertainty is provided for any of the predictions given in the results (lines 277-303 and figures 4-5).

We agree the term validation is inappropriate. Instead, we have opted to adjust the wording of this section, have improved the parameterisation of the overall predictions and have included the variation in the predicted change in prevalence using the maximum and minimum performance of pirimiphos-methyl products. We use the seasonal profile for the administration unit of Kagera, Tanzania rather than re-fit to rainfall for the respective years of these RCTs. We now use these figures to illustrate the utility of the IRS fitting rather than consider these as validation.

To address the issues of uncertainty, we now present a statistical analysis of the initial data to show differences between IRS products, mosquito species, wall substrates and hut types (restricting this to East or West African hut design) (new Fig. 1 and see Supplementary figures S2 – S5). The limited data for time series analyses do not allow us to understand how these covariates might alter the duration of impact. Instead, to present the uncertainty, we now used the best and worst performing experimental hut studies for each IRS product and then generate estimates that span this range, rather than presenting a single average estimate. We add a minimum and maximum tab for each IRS product (new Fig. 2, Supplementary Figure 8) to demonstrate the effect of this range on IRS predicted performance. We also make it clearer that there is substantial uncertainty and limitations to the analysis within the text [lines 462 – 479 and throughout results / methods sections].

For the location-specific predictions, we use site-specific parameters to capture the seasonality and mosquito species compositions within administration sites. We can provide these site-specific parameter estimates if requested to help with interpretation of these estimates. We now show this in Fig. 4 by adding dashed lines to indicate the predictions from the different Actellic or Bendiocarb studies. We overlay these onto the polygons showing the maximum and minimum range predicted

and the uncertainty of other model parameters around the mean parameter fit for the IRS product in question.

Minor comments

Lines 49-53. Ref. 1 showed that the coverage of ITNs and IRS that was achieved in Africa during 2000-2015 had a larger impact than the coverage of ACTs that had been achieved during the same period, that is not the same as showing that ITN/IRS are more effective tools than ACT. The results of refs 2-3 are stated as fact but they are actually predictions.

Adjusted: “The majority of this decline is *thought to be* through LLIN use, though nets alone will ... “

I recommend getting someone to proof read the Introduction to ensure it flows well and that the use of parentheses within parentheses doesn't render sentences hard to read. For example, within lines 60-77 the sentences don't always follow on from each other in an obvious way and the final sentence seems like an outlier. The Discussion is much better.

Thank you, we have taken this advice and asked colleagues to read-through for flow. We have re-written the noted section [lines 65 – 83] and other sections of the introduction.

The search for studies that contributed to the meta-analysis was conducted on 25 May 2015, which is over three years ago. Given the lack of data, it's surprising the authors didn't update this search.

The meta-analysis has been updated to 27th July 2018. Supplementary Table 1 highlights which studies are included in analysis 1, 2, 3a and 3b and the reasons why data are excluded. Supplementary Figure 1 summarises the meta-analysis for repeatability.

Lines 462-465. As Nature Communications has a broad authorship I recommend defining 'exophily' (not just the mathematical/experimental hut definition). It may also be worth including a brief explanation of how an experimental hut trial works.

Added: “Exophily is the propensity for mosquitoes to rest outdoors after feeding which can diminish the impact of IRS.”

Lines 528-533. Be careful about stating WHO tube assays, etc assess the 'level' or 'severity' of resistance. Unless concentration is varied (which is rarely the case) they only measure the proportion of mosquitos from a population sample killed by a standard dose.

Adjusted to: “Discriminating dose bioassays (WHO tube assay, WHO cone assay, CDC bottle assay) are a practical option for control programmes *to assess the proportion of the mosquito population that are killed by a standard dose. The assumption is made that the inverse of this proportion, i.e. those mosquitoes surviving in the presence of the standard dose of insecticide, is representative of the level of insecticide resistance in the mosquito population.*” [Lines 684 – 687]

Lines 531-541. A recent study (Hancock 2018 PNAS 115:5938) provides further justification for pooling pyrethroid data although this is still a limitation of the study as some differences are found.

This reference has been added. Thank you

Lines 548-561. The species composition in Table 4 doesn't match that reported by Protopopoff but it's not clear why. What 'level' of resistance was assumed for *funestus*?

Apologies – this was my error and has been corrected. The model has been re-fitted and adjusted to include the uncertainty in IRS performance. We are now better able to match this study (Fig. 3). But we also improve this evidence by presenting the predictions using parameter estimates determined by individual studies too. The footnotes to Table 3 note our assumptions on *An. arabiensis*.

Lines 383-4. There is evidence to support the statement about resistance varying between species, e.g. Ondeto (2017) P&V 10:429, etc, etc.

This reference has been added. Thank you

Lines 385-387. Were data from West African hut designs excluded from all parts of the study including the information that went into the simulations across the whole of Africa? This really highlights the limitation of validating part of the work only, using data from one (eastern) location and then simulating transmission across many scenarios. I didn't spot any mention of this exclusion in the Methods or the Results, just the Discussion.

All studies were included within the binomial logistic regression analysis using the initial data (results for mosquito counts within 2 months of the start of an experimental hut trial) (further methods and results for these are presented in Supplementary Methods 1).

Principally West African hut studies were included (Table 2) for the times series analysis given the clear discrepancy in mortality rates for Ifakara huts.

Only studies with either pyrethroid time series and an initial measure of mortality (within 2 months of spraying huts) or a measure of both initial hut mortality *and* a measure of WHO tube bioassay mortality in a standard dose assay could be included for the analysis of pyrethroid resistance.

We have provided these studies and where each were included in Supplementary Table 1 and each dataset (excluding unpublished data) in supplementary Data 1, to clarify this query. We have also rewritten much of the results and methods to clarify which data were used for which analysis.

It needs to be clear when the results are presented which come from a model that assumes moderate (30%) endemicity and no prior interventions, and which come from a model that uses data for seasonal profiles and data on past scale-up of IRS and LLIN.

This has been made clearer with the re-structured results section.

Check the meaning of principle vs principal.

Thank you. Corrected to: "Whilst, pyrethroids are the principal insecticide used in nets, new IRS products have been recently launched."

Reviewer #3:

NCOMMS-18-18452

The authors have conducted a meta-analysis of experimental huts trials to characterize the efficacy of different IRS products on anophelines. These data have been used to parametrize an established model and predict the impact of different IRS product classes across a range of eco-epidemiological situations with varying LLIN coverage and pyrethroid resistance. The results are compared against two published trial for validation with relative success.

This allows for efficacy and cost assumptions in different scenarios which can be an asset for NMCP managers and those involved in drafting national strategic plans. Donors and implementers could also benefit. Assumptions are well laid out and limitations are reasonably explained. The conclusions are supported by the results.

There are however some methodological issues that require clarification.

MAJOR

The authors refer to the background review as a “systematic meta-analysis” (line 440). PRISMA guidelines are mentioned in line 439.

The PRISMA checklist (<http://prisma-statement.org/prismastatement/Checklist.aspx>) however has not been followed. Some examples are:

- The title does not identify the report as a systematic review or meta-analysis (or both)
- There is no reference to a protocol or registered methodology
- The description of search methods and assessment of title/abstracts is limited

We have re-done the meta-analysis and present the systematic review in Supplementary methods 1 and Supplementary Figure 1. We have adjusted the title according to these guidelines. We originally tried to register the review on PROSPERO but were rejected. The email response from that system states “Reviews that have progressed beyond the point of completing data extraction are not eligible for inclusion in PROSPERO. The aim of the register is to capture information at the protocol stage of a review [16th August 2016]”. Unfortunately, this restricts us providing a registration number for the review. In updating the analysis, we have identified 41 studies to include so this was a very useful exercise.

MINOR

Introduction:

There is no mention of residual transmission as a challenge although the mosquito behavior leading to it is referred to in lines 373-375

We have deliberately not focused on residual transmission because this is a big topic for the assessment of indoor vector control that is beyond the scope of the current work. The paper is already substantial, and residual transmission will require much discussion ultimately. The model is able to account for residual transmission provided assumptions are made on the proportion of mosquitoes that attempt to blood-feed indoors and on people. In the current work these are kept

constant. We have addressed this as a separate piece of work that is still in preparation and opted not to add a commentary here given the length of the submission.

Authors should mention and include in the references the WHO guidance for countries on combining indoor residual spraying and long-lasting insecticidal nets.

Added (reference 4)

Lines 116-119: should mention and include in references the WHO information note (2017) on the evaluation process for vector control products.

Added (reference 27)

Lines 76-77: the phrase is incomplete “LLINs still provide personal protection by acting as a physical barrier and invoking minimal killing and some repellence”. In presence of resistance I presume?

Adjusted to: “LLINs still provide personal protection by acting as a physical barrier and invoking minimal killing and some repellence *in the presence of resistance*”

Results:

Lines 161-162: please elaborate further in the discussion on the potential implications of combining the results of *An. gambiae* s.l. and *An. funestus* as well as different hut designs and substrates.

We have included much more uncertainty in our model predictions (following suggestions from reviewer 1), in doing so we hope to cover some of the challenges of having minimal data to consider species and hut designs distinctly. We add a distinct section to the discussion to note clearly the limitations of the analyses (lines 462 – 479).

Discussion:

Lines 420: 80% coverage might not be financially or operationally achievable everywhere or with all chemistries. Different chemistries have different effect on the walls (staining) and different smells which directly affects acceptability.

Added and adjusted to: “These estimates are determined using IRS at 80% cover which may not be financially achievable everywhere *or with all chemistries. Different chemistries have different effects on different wall surfaces*⁶ *as well as contrasting smells or propensity to leave stains which affects acceptability*⁸. The present analysis assessing the potential impact of IRS at different levels of pyrethroid resistance *can contribute to decision making.*”

1. Thawer, N. G. *et al.* Use of insecticide quantification kits to investigate the quality of spraying and decay rate of bendiocarb on different wall surfaces in Kagera region , Tanzania. *Parasit. Vectors* **8**, 242–251 (2015).
2. Protopopoff, N. *et al.* Effectiveness of a long-lasting piperonyl butoxide-treated insecticidal net and indoor residual spray interventions, separately and together, against malaria transmitted by pyrethroid-resistant mosquitoes: a cluster, randomised controlled, two-by-two fact. *Lancet* **391**, 1577–1588 (2018).

3. Agossa, F. R. *et al.* Efficacy of various insecticides recommended for indoor residual spraying: pirimiphos methyl, potential alternative to bendiocarb for pyrethroid resistance management in Benin, West Africa. *Trans. R. Soc. Trop. Med. Hyg.* **108**, 84–91 (2014).
4. Rowland, M. *et al.* A new long-lasting indoor residual formulation of the organophosphate insecticide pirimiphos methyl for prolonged control of pyrethroid-resistant mosquitoes: an experimental hut trial in Benin. *PLoS One* **8**, e69516 (2013).
5. Tchicaya, E. S. *et al.* Micro-encapsulated pirimiphos-methyl shows high insecticidal efficacy and long residual activity against pyrethroid-resistant malaria vectors in central Côte d'Ivoire. *Malar. J.* **13**, 332 (2014).
6. Oxborough, R. M. *et al.* Long-lasting control of *Anopheles arabiensis* by a single spray application of micro-encapsulated pirimiphos-methyl (Actellic® 300 CS). *Malar. J.* **13**, 37 (2014).
7. Darriet, F. *et al.* Impact de la résistance aux pyréthrinoïdes sur l'efficacité des moustiquaires imprégnées dans la prévention du paludisme: résultats des essais en cases expérimentales avec la deltaméthrine SC. *Bull Socc Pathol Exot* **93**, 131–134 (2000).
8. Kaufman, M. R., Rweyemamu, D., Koenker, H. & Macha, J. "My children and I will no longer suffer from malaria": a qualitative study of the acceptance and rejection of indoor residual spraying to prevent malaria in Tanzania. *Malar. J.* **11**, 220 (2012).

We thank all reviewers for very useful and constructive advice.

REVIEWERS' COMMENTS:

Reviewer #1 (Remarks to the Author):

Thanks to the authors for their thoughtful edits and additions in response to the reviewer comments. This is an improved body of work that I am comfortable recommending for publication.

Reviewer #2 (Remarks to the Author):

The manuscript is much improved and I have no further questions or concerns.

Reviewer #3 (Remarks to the Author):

I have re-reviewed the manuscript which has retained its originality and improved noticeably after assessing reviewer's comments. All my concerns have been fully addressed.